# Dysbindin links presynaptic proteasome function to homeostatic recruitment of low release probability vesicles

Corinna Wentzel[1], Igor Delvendahl[1], Sebastian Sydlik[1,2], Oleg Georgiev[1] & Martin Müller[1,2,3]

Here we explore the relationship between presynaptic homeostatic plasticity and proteasome function at the *Drosophila* neuromuscular junction. First, we demonstrate that the induction of homeostatic plasticity is blocked after presynaptic proteasome perturbation. Proteasome inhibition potentiates release under baseline conditions but not during homeostatic plasticity, suggesting that proteasomal degradation and homeostatic plasticity modulate a common pool of vesicles. The vesicles that are regulated by proteasome function and recruited during homeostatic plasticity are highly EGTA sensitive, implying looser $Ca^{2+}$ influx-release coupling. Similar to homeostatic plasticity, proteasome perturbation enhances presynaptic $Ca^{2+}$ influx, readily-releasable vesicle pool size, and does not potentiate release after loss of specific homeostatic plasticity genes, including the schizophrenia-susceptibility gene *dysbindin*. Finally, we provide genetic evidence that Dysbindin levels regulate the access to EGTA-sensitive vesicles. Together, our data suggest that presynaptic protein degradation opposes the release of low-release probability vesicles that are potentiated during homeostatic plasticity and whose access is controlled by *dysbindin*.

---

[1] Institute of Molecular Life Sciences, University of Zurich, Winterthurerstrasse 190, 8057 Zurich, Switzerland. [2] Zurich Ph.D. Program in Molecular Life Sciences, Winterthurerstrasse 190, Zurich, 8057, Switzerland. [3] Neuroscience Center Zurich, Winterthurerstrasse 190, Zurich, 8057, Switzerland. Correspondence and requests for materials should be addressed to M.Mül. (email: Martin.Mueller@imls.uzh.ch)

Most synapses function robustly within neural circuits throughout the lifetime of an animal. However, the proteins determining synaptic activity are turned over on relatively rapid time scales of hours to weeks[1–3]. Thus, individual synapses are continuously 'rebuilding' themselves. It is therefore remarkable that robust neural function can be maintained at all. Work during the last decades has revealed that synaptic transmission is actively stabilized by evolutionarily conserved homeostatic signaling systems, but the underlying molecular mechanisms are largely unknown[4–7].

The *Drosophila* neuromuscular junction (NMJ) has emerged as a powerful model synapse (note that we use 'NMJ' and 'synapse' interchangeably) to unravel the genetic basis of presynaptic homeostatic plasticity (PHP)[8]. At this synapse, postsynaptic glutamate receptor perturbation induces a homeostatic increase in neurotransmitter release that precisely restores postsynaptic depolarization in response to action-potential (AP) stimulation[9,10]. An electrophysiology-based genetic screen has implicated several genes in this form of synaptic plasticity[11–18]. All of the identified genes are evolutionarily conserved to humans, and loss-of-function mutations in these genes block PHP. Nevertheless, it has remained unclear how most of the identified genes participate in homeostatic signaling.

One of the genes that is required for PHP in *Drosophila* is *dysbindin*[11], a gene linked to schizophrenia in humans (*DTNBP1*[19,20]). There is genetic evidence that *dysbindin* functions in concert with *snapin* and SNAP-25 during PHP[21]. Recent genetic data suggest that *dysbindin*-dependent homeostatic regulation of release involves the BLOC-1 complex member *Blos1*[22] and the Arp2/3 actin polymerization complex[23]. Moreover, the role of *dysbindin* in PHP appears to be independent of presynaptic $Ca^{2+}$ influx or major changes in synapse development[11]. However, besides these genetic data little is known about how *dysbindin* is involved in PHP.

At the *Drosophila* NMJ, PHP can be rapidly induced within minutes in the presence of the protein synthesis inhibitor cyclohexamide[12], suggesting that the acute induction of PHP does not require synthesis of new proteins. In contrast, protein degradation has not been studied in the context of PHP at this synapse. The ubiquitin-proteasome system (UPS) is a major protein degradation pathway[24]. At the *Drosophila* NMJ it has been shown that all components of the UPS are present at presynaptic terminals and that acute proteasome inhibition causes a rapid strengthening of neurotransmission[3]. There is accumulating evidence for links between neural activity and UPS-mediated degradation of presynaptic proteins in mice and rats[25–30]. However, only two presynaptic proteins—Rab3-interacting molecule (RIM[25,26,30]) and Dunc-13/munc-13[3]—as well as one E3-ligase (SCRAPPER[30]) have so far been implicated in UPS-dependent control of presynaptic protein turnover and release. Thus, the molecular pathways underlying the regulation of presynaptic release through protein degradation remain enigmatic.

PHP at the *Drosophila* NMJ requires high-release probability ($p_r$) vesicles that are 'tightly coupled' to $Ca^{2+}$ channels through *rim-binding protein*[16]. The distance between $Ca^{2+}$ channels and $Ca^{2+}$ sensors of exocytosis, typically referred to as 'coupling distance', is a major factor determining the $p_r$ of synaptic vesicles[31–34]. However, despite their implication in short-term plasticity, little is known about how vesicles that are differentially coupled to $Ca^{2+}$ influx are modulated during synaptic plasticity.

Here we explore the relationship between PHP and presynaptic proteasome function at the *Drosophila* NMJ and provide evidence for links between presynaptic protein degradation and homeostatic potentiation of loosely-coupled synaptic vesicles.

## Results

**PHP requires presynaptic proteasome function.** To investigate potential links between protein degradation and PHP induced by postsynaptic glutamate receptor perturbation, we probed the effects of acute pharmacological proteasome inhibition on PHP at the *Drosophila* NMJ (Fig. 1). First, we acutely perturbed proteasome function by application of the proteasome inhibitor lactacystin (100 μM for 15 min; Supplementary Fig. 1), which has been demonstrated to interfere with synaptic proteasome function at the *Drosophila* NMJ before[3]. Using this manipulation, we asked if protein degradation is required for the acute induction of PHP in response to pharmacological glutamate receptor perturbation[12]. Application of sub-saturating concentrations of the glutamate receptor antagonist philanthotoxin-433 (PhTX, 20 μM for 10 min) led to a significant decrease in miniature excitatory postsynaptic potential (mEPSP) amplitude at both control and lactacystin-treated synapses (Fig. 1b, left). At control synapses, the decrease in mEPSP amplitude in the presence of PhTX was accompanied by a significant increase in neurotransmitter release ('quantal content'=EPSP amplitude/mEPSP amplitude; Fig. 1b, c, right), thereby restoring AP-evoked EPSP amplitudes to baseline levels (Fig. 1b, middle), as expected from previous work[9,12]. By contrast, synapses that had previously been incubated with lactacystin had elevated baseline EPSP amplitudes (Fig. 1b, middle; see below), and PHP was completely blocked, as shown by similar quantal content values for lactacystin-treated synapses in the absence and presence of PhTX (Fig. 1b, right). As a consequence, EPSP amplitudes were significantly decreased after PhTX treatment (Fig. 1b, middle), to a similar degree as mEPSP amplitudes (Fig. 1b, left). Similar results were obtained with the proteasome inhibitor MG-132 (Fig. 1b). These data demonstrate that proteasome function is required for the acute induction of PHP.

We also observed that lactacystin treatment potentiates quantal content (Fig. 1b, right) in a concentration-dependent and incubation time-dependent manner under baseline conditions without affecting mEPSP frequency (Supplementary Fig. 1, 2a and b), consistent with previous work[3]. The block of PHP after lactacystin treatment could be due to saturation of quantal content. Although we cannot fully exclude a ceiling effect, we consider this unlikely because release was not saturated after lactacystin application under our recording conditions (quantal content ~35 vs. ~50 after PhTX treatment; Fig. 1b), and because PHP was completely blocked after proteasome perturbation at elevated extracellular $Ca^{2+}$ concentration (1 mM; see below). Together, our results suggest that proteasome function opposes the release of synaptic vesicles that are required for PHP.

The effects of lactacystin could arise from presynaptic and/or postsynaptic proteasome impairment. To distinguish between these two possibilities, we took advantage of the *Gal4/UAS* expression system to specifically perturb either presynaptic or postsynaptic proteasome function (neuronal driver line *elav^{c155}*-Gal4, and muscle-specific *24B-Gal4* driver line, respectively). Using these *Gal4* lines, we overexpressed one or two dominant temperature-sensitive mutant proteasome subunits (*UAS-DTS5* and *UAS-DTS7*; see Methods), which were demonstrated to impair proteasome function in *Drosophila* because of mutations in β-subunits of the 20S proteasome[35]. Under our experimental conditions (expression at the permissive temperature, 25 °C, throughout development), *DTS* expression is thought to result in a hypomorphic phenotype, i.e., a perturbation, but not complete loss of proteasome function[35]. We first assessed if presynaptic overexpression of two *DTS* copies—henceforth referred to as '*DTS^{pre}*'—affects the acute induction of PHP upon PhTX treatment (Fig. 1a, c). Application of PhTX to *DTS^{pre}* mutants

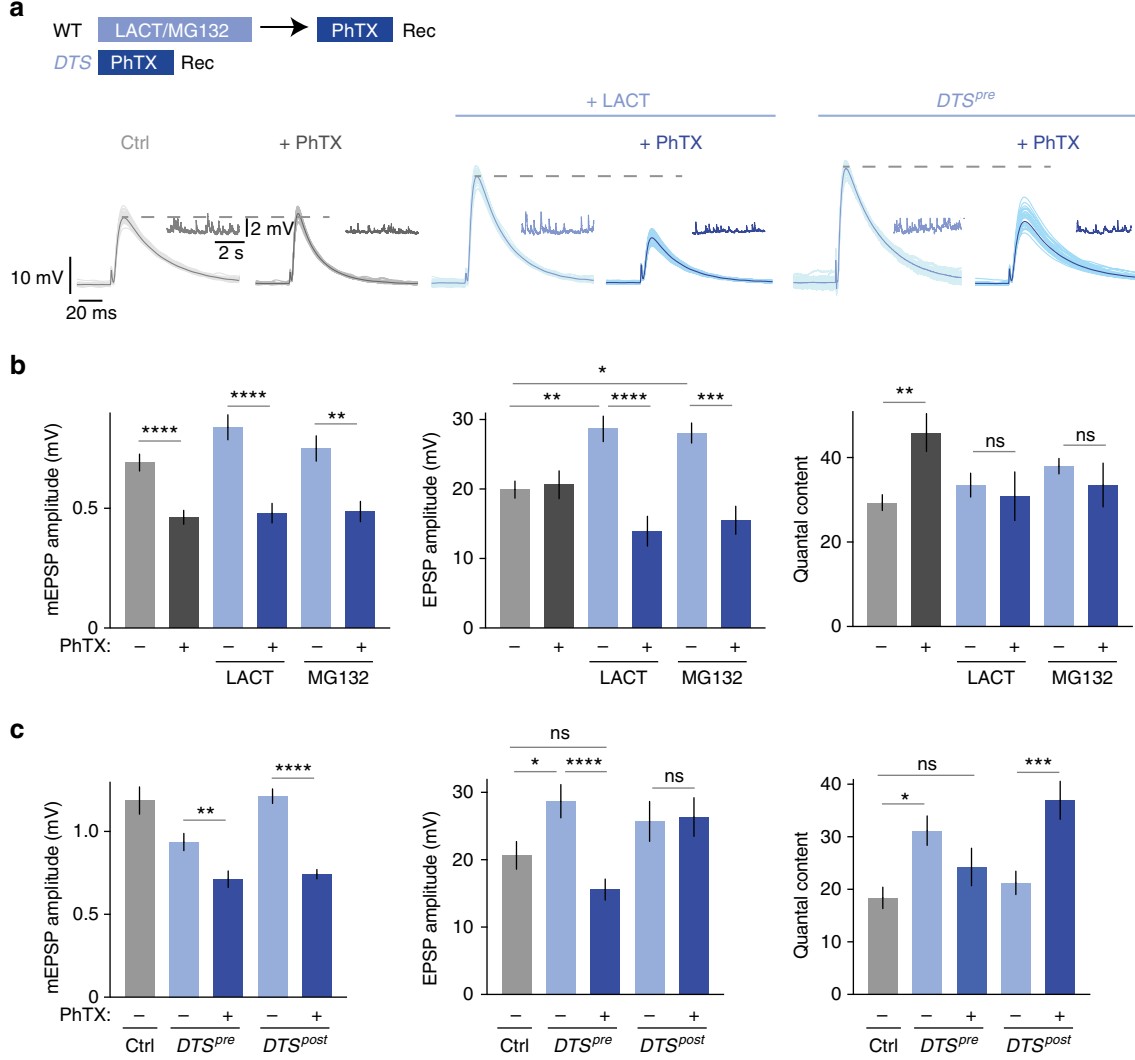

**Fig. 1** Presynaptic proteasome perturbation blocks presynaptic homeostatic plasticity. **a** Scheme for treatment of NMJs and representative EPSP and mEPSP traces (Rec: recording). Wild-type NMJs were incubated with saline, 100 μM lactacystin for 15 min or 50 μM MG-132 for 15 min, followed by wash and treatment with saline or 20 μM PhTX for 10 min at 0.3 mM $[Ca^{2+}]_e$. *DTS*-mutant synapses were treated with saline or 20 μM PhTX for 10 min at 0.3 mM $[Ca^{2+}]_e$. A total of 30 sweeps are shown per cell, with the average displayed in darker color. **b** Quantification of mEPSP amplitude, EPSP amplitude and quantal content for wild-type synapses treated as described in (**a**). PhTX treatment induced an increase in quantal content (right). Note that PhTX application after lactacystin or MG-132 treatment completely blocks the homeostatic increase in quantal content (right). Mean ± s.e.m.; $n \geq 6$ cells; *$p < 0.05$; **$p < 0.001$; ***$p < 0.0001$; ****$p < 0.00001$; ns: not significant; ANOVA and Tukey's multiple comparison tests for comparison of EPSP amplitudes, Student's *t*-test for pairwise comparison of mEPSP amplitude and quantal content in PhTX-treated vs. non-PhTX-treated cells. **c** Quantification of mEPSP amplitude, EPSP amplitude and quantal content of control synapses (*elav*[C155]-Gal4 and 24B-Gal4) and synapses expressing mutant proteasome subunits presynaptically (*elav*[C155]-Gal4;UAS-DTS5/+;UAS-DTS7/+; '*DTS*[pre]') or postsynaptically (*UAS-DTS5/+;UAS-DTS7/24B-Gal4; '*DTS*[post]'). Synapses were treated with 20 μM PhTX or saline for 10 min at 0.3 mM $[Ca^{2+}]_e$. Note that there is no increase in quantal content upon PhTX treatment after genetic presynaptic proteasome perturbation (*DTS*[pre]), indicating a defect in presynaptic homeostatic plasticity. PhTX treatment after postsynaptic proteasome perturbation potentiates quantal content (right), demonstrating that postsynaptic proteasome mutants do not have a defect in homeostatic plasticity. Mean ± s.e.m.; $n \geq 10$ cells; *$p < 0.05$; **$p < 0.001$; ***$p < 0.0001$; ****$p < 0.00001$; ns: not significant; ANOVA and Tukey's multiple comparison tests

reduced mEPSP amplitudes (Fig. 1c, left), but failed to increase quantal content with respect to untreated *DTS* mutants (Fig. 1c, right), resulting in significantly smaller EPSP amplitudes compared to untreated mutants (Fig. 1c, middle). These data demonstrate that the presynaptic proteasome is required for PHP. We also detected a significant increase in EPSP amplitude and quantal content in *DTS*[pre] mutants under baseline conditions (Fig. 1a, c) that was similar to the lactacystin-induced increase in release (Fig. 1b).

Moreover, while observing a significant increase in release after presynaptic expression of two *DTS* copies, there was no apparent change in synaptic transmission upon expressing one *DTS* copy

(Supplementary Fig. 2c and d). This implies a dose-dependent effect of *DTS* expression on release, similar to lactacystin treatment (Supplementary Fig. 1). Again, release was not saturated at *DTS*[pre]-mutant synapses in the absence of PhTX (Fig. 1c), suggesting that presynaptic genetic proteasome inhibition unlikely occluded PHP, similar to pharmacological proteasome inhibition.

By contrast, PhTX treatment of postsynaptic *DTS*-mutant synapses ('*DTS*[post]', Fig. 1c) enhanced quantal content with respect to untreated mutants (Fig. 1c, right), thereby restoring EPSP amplitudes to baseline levels (Fig. 1c, middle). Hence, postsynaptic proteasome function is not required for the acute

induction of PHP, underlining the relevance of the presynaptic proteasome in this form of synaptic plasticity.

A recent study demonstrated reduced presynaptic proteasome number and mobility because of a defect in proteasome trafficking in mutants lacking *cut-up* (*ctp*)[36], a gene encoding the *Drosophila* homolog of the dynein light chain subunit LC8. Interestingly, we revealed that loss of *ctp* disrupts PHP (Supplementary Fig. 2e and f), further supporting our conclusion that presynaptic proteasome function is required for PHP.

In addition, we tested if the enhancement in synaptic strength observed after lactacystin application goes along with an increase in ubiquitinated protein levels at the synapse (Supplementary Fig. 1d–g). To assay synaptic levels of ubiquitin-tagged proteins, we utilized an antibody (FK2) against mono- and polyubiquiti-nated proteins but not free ubiquitin[37]. Lactacystin treatment induced a pronounced increase in FK2-fluorescence intensity throughout the synapse and the postsynaptic muscle cell (Supplementary Fig. 1d–g). Since our physiology data suggest a presynaptic function, we quantified FK2-fluorescence intensity in the presynaptic compartment using a mask based on the neuronal membrane marker horseradish peroxidase (HRP, see Methods[38]) and detected a marked increase in FK2 fluorescence without apparent changes in HRP fluorescence upon lactacystin treatment (Supplementary Fig. 1e–g). These data demonstrate that the increase in neurotransmitter release is correlated with a rapid increase in presynaptic ubiquitinated proteins.

**Proteasome perturbation does not enhance release during PHP**. We demonstrated that lactacystin application enhances release under baseline conditions, whereas subsequent PhTX treatment fails to induce PHP. We next reversed the order of the experimental perturbations and probed the effects of lactacystin treatment on release after inducing PHP by PhTX incubation (Fig. 2a). An increase in release upon proteasome impairment after the induction of PHP would imply that both manipulations activate different cellular pathways resulting in additive effects. By contrast, an absence of an effect would indicate that both per-turbations are non-additive. As shown above (Fig. 1), lactacystin treatment significantly increased EPSP amplitudes and quantal content at control synapses that were not treated with PhTX (Fig. 2a, b). However, we did not detect changes in EPSP amplitude or quantal content upon lactacystin treatment if synapses had been previously incubated with PhTX as compared to synapses that were treated with PhTX alone (Fig. 2a, b, middle, right). These results are consistent with the idea that the increase in release induced by either PhTX incubation or lactacystin treatment employ similar cellular mechanisms and are non-additive.

At the *Drosophila* NMJ, PHP can be either acutely induced by PhTX application or chronically expressed through genetic ablation of the glutamate receptor subunit *GluRIIA* (*GluR-IIA^SP16*)[10]. We next investigated the relationship between proteasome function and the sustained expression of PHP (Fig. 2a, b). Similar to wild-type synapses that were treated with PhTX (Fig. 2a, b), *GluRIIA* mutants did not display significant changes in EPSP amplitude or quantal content after lactacystin treatment compared to saline-treated *GluRIIA* mutants (Fig. 2b). This observation implies that the mechanisms underlying acute and sustained PHP expression are required to potentiate release upon acute proteasome impairment.

Next, we generated transgenic flies that are mutant for *GluRIIA* and overexpress *DTS* presynaptically ('*GluRIIA, DTS^pre*'; flies were kept at the permissive temperature throughout develop-ment) to investigate presynaptic proteasome function in the context of sustained PHP. Similar to lactacystin application, genetic presynaptic proteasome inhibition caused no apparent changes in EPSP amplitude or quantal content compared to *GluRIIA* mutants alone (Fig. 2c, d). This suggests that sustained PHP expression involves mechanisms that are required to potentiate release upon proteasome inhibition. By extension, the fact that pharmacological or genetic proteasome perturbation do not induce apparent changes in synaptic physiology after the induction or sustained expression of PHP suggests that the effects of proteasome impairment on release are specific to homeostatic plasticity.

Next, we tested if synapses that undergo PHP exhibit changes in ubiquitinated protein levels using the FK2 antibody. We did not detect significant changes in FK2-fluorescence intensity upon PhTX treatment (Supplementary Fig. 3), but there was a consistent and significant increase in FK2-fluorescence intensity in the presynaptic mask of *GluRIIA*-mutant synapses compared to wild-type controls (Fig. 2e). Even if we cannot rule out that the observed increase in FK2-fluorescence intensity at *GluRIIA*-mutant synapses is due to increased protein expression, these data suggest that the sustained expression of PHP is correlated with increased synaptic levels of ubiquitin-tagged proteins. The observation of no apparent changes in FK2-fluorescence intensity after PhTX incubation suggests that we either cannot resolve potential differences, or that non-ubiquitinated protein levels may be altered during acute homeostatic plasticity. Alternatively, different mechanisms may underlie the acute induction and the sustained expression of PHP.

**Unaltered synapse morphology after proteasome perturbation**. The defect in PHP in presynaptic *DTS* mutants could be caused by impaired synapse development. Therefore, we investigated synaptic morphology in flies overexpressing two *DTS* copies ('*DTS^pre*') neuronally throughout development (Fig. 3). Immu-nostainings of NMJs using antibodies detecting neuronal mem-branes ('aHRP', anti-horseradish peroxidase), presynaptic active zones ('aBrp', anti-Bruchpilot) and the postsynaptic reticulum ('aDLG', anti-discs large) revealed no apparent changes in synapse size (synaptic span, HRP area and DLG area), active zone number (Brp puncta number), active zone density (Brp puncta/ HRP area and Brp puncta/DLG area), active zone area (Brp area) or Brp intensity between *DTS^pre* mutants and controls (Fig. 3b, Supplementary Fig. 4). Moreover, there were no apparent dif-ferences in glutamate receptor field number, size or glutamate receptor levels between *DTS^pre* mutants and wild types at con-focal resolution (Supplementary Fig. 4). Thus, the disruption of PHP is unlikely a secondary consequence of impaired anatomical development or reduced active zone number at presynaptic *DTS*-mutant NMJs.

**Presynaptic proteasome perturbation potentiates EPSC charge**. Next, we examined PHP after presynaptic *DTS* expression using two-electrode voltage-clamp (TEVC) analysis at increased $[Ca^{2+}]_e$ (Fig. 4; 1 mM $[Ca^{2+}]_e$). PhTX treatment significantly decreased mEPSC amplitude and mEPSC charge at control and *DTS^pre*-mutant synapses (Fig. 4a, d). At control synapses, the PhTX-induced decrease in mEPSC amplitude or mEPSC charge resulted in a significant increase in quantal content (Fig. 4e, gray data), which restored EPSC amplitudes and EPSC charge towards baseline values (Fig. 4b, gray data). By contrast, PhTX application did not potentiate quantal content in *DTS^pre* mutants (Fig. 4e, blue data), so that EPSC amplitude and EPSC charge remained significantly smaller than at untreated mutant synapses (Fig. 4b, blue data). Thus, presynaptic proteasome function is also required for PHP at elevated $[Ca^{2+}]_e$.

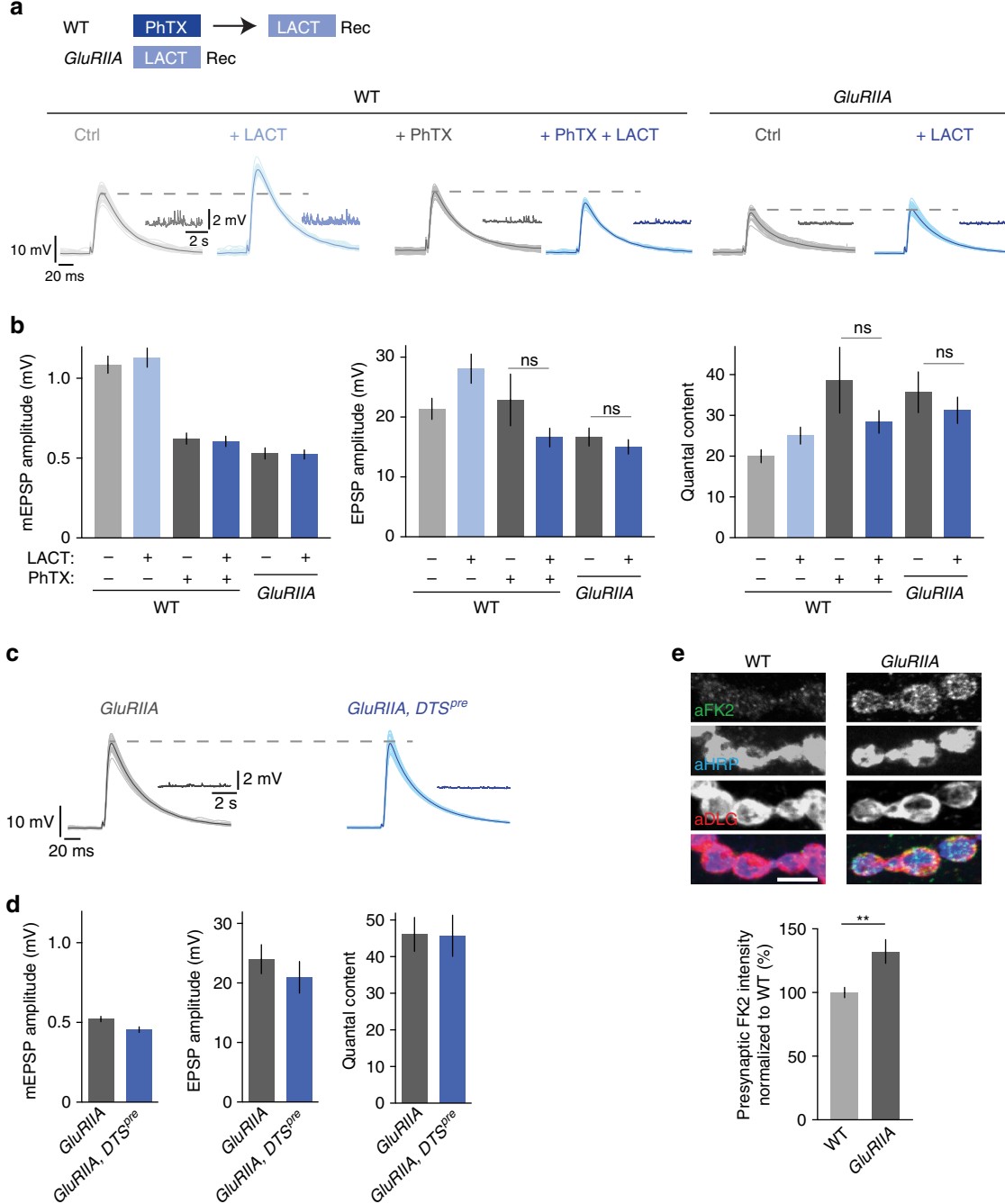

**Fig. 2** Proteasome perturbation does not potentiate release after acute induction or sustained expression of PHP. **a** Scheme for treatment of NMJs and representative EPSP and mEPSP traces. Wild-type NMJs were incubated with saline or 20 μM PhTX for 10 min, followed by a wash and treatment with saline or 100 μM lactacystin for 30 min at 0.3 mM $[Ca^{2+}]_e$. *GluRIIA*-mutant synapses were treated with saline or 100 μM lactacystin for 30 min at 0.3 mM $[Ca^{2+}]_e$. **b** Quantification of EPSP amplitude, mEPSP amplitude and quantal content for larvae treated as described in (**a**). Lactacystin application to larvae treated with PhTX or to *GluRIIA*-mutant larvae has no effect on EPSP amplitude or quantal content. Mean ± s.e.m.; $n \geq 10$ cells; ns: not significant; Student's *t*-test. **c** Representative EPSP and mEPSP traces of *GluRIIA*-mutant synapses and *GluRIIA,DTS^pre* double-mutant synapses. **d** Quantification of mEPSP amplitude, EPSP amplitude and quantal content of the genotypes described in (**c**). Presynaptic proteasome perturbation does not increase EPSP amplitude or quantal content in *GluRIIA*-mutant synapses. Mean ± s.e.m.; $n \geq 7$ cells. **e**) Representative NMJs stained for mono- and polyubiquitinated proteins (aFK2), neuronal membrane (aHRP, horseradish peroxidase), and postsynaptic reticulum (aDLG, discs large) of wild-type and *GluRIIA*-mutant larvae (scale bar, 5 μm). Quantification of FK2-fluorescence intensity in the HRP mask (presynapse) suggests that *GluRIIA*-mutant NMJs have higher levels of mono- and polyubiquitinated proteins compared to wild-type NMJs. Mean ± s.e.m.; $n = 24$ NMJs; **$p < 0.001$; Student's *t*-test

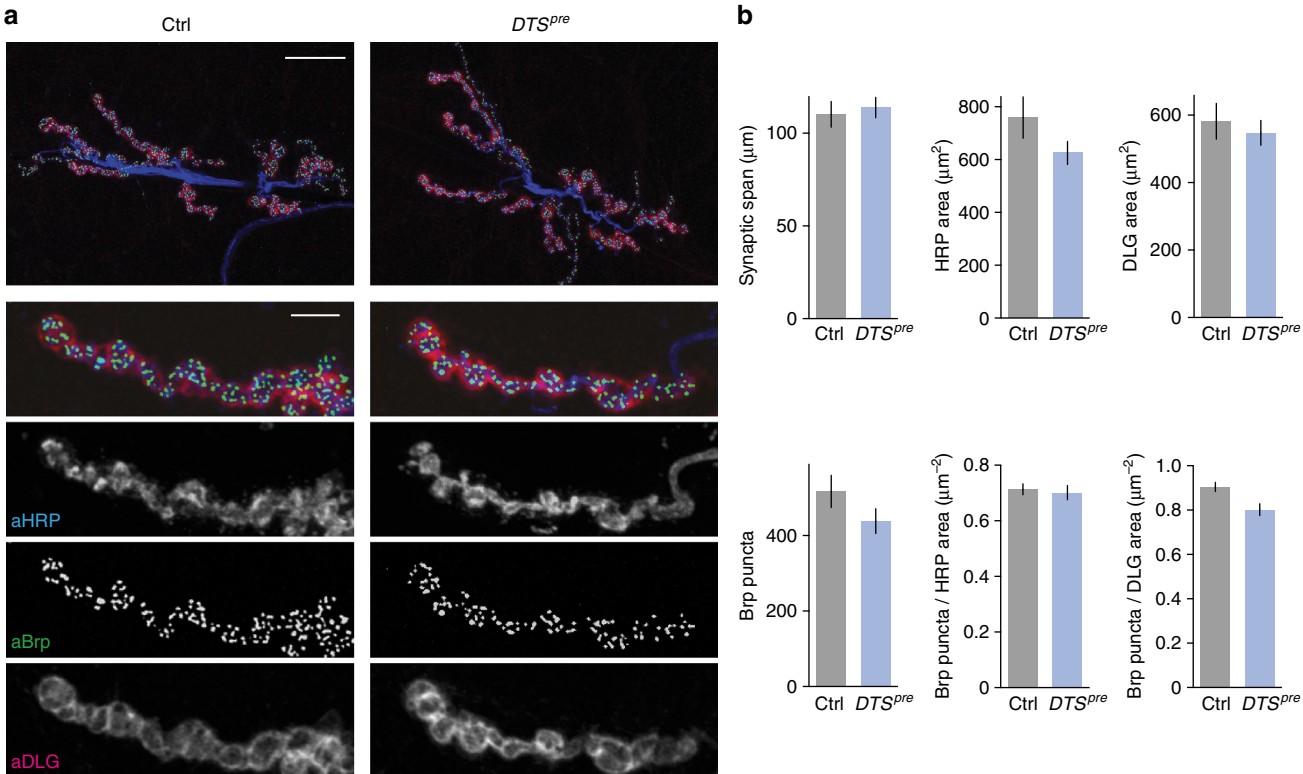

**Fig. 3** Presynaptic overexpression of the mutant proteasome subunit *DTS* does not induce apparent changes in synapse morphology. **a** Immunostainings of a control NMJ (*elav*[C155]-*Gal4*) and an NMJ expressing two copies of the mutant proteasome subunit *DTS5* presynaptically (*elav*[C155]-*Gal4;UAS-DTS5*; '*DTS*[pre]'). Larvae were kept at the permissive temperature (25 °C) throughout development. NMJs were stained for neuronal membrane (aHRP), postsynaptic reticulum (aDLG) and the active zone marker Bruchpilot (aBrp); scale bar overview = 20 μm; scale bar inset = 5 μm. **b** Quantification of NMJ parameters for the two genotypes shown in (**a**). Presynaptic *DTS* overexpression does not result in apparent changes in synapse morphology. Mean ± s.e. m.; *n* = 24 NMJs; Student's *t*-test

When comparing EPSCs between *DTS*[pre] mutants and controls in the absence of PhTX, we noted that presynaptic *DTS* overexpression induced an increase in EPSC amplitude as well as a slowing of EPSC decay kinetics (Fig. 4b, c; light gray and light blue data). Together, both effects translate into a significant potentiation of EPSC charge in *DTS*[pre] mutants (Fig. 4b). It is worth noting that changes in EPSC charge are of physiological relevance at this synapse because they translate into changes in EPSP amplitudes due to the relatively slow membrane time constant of the postsynaptic muscle cell ($\tau \approx 40$ ms)[39]. The observed increase in EPSC charge upon presynaptic *DTS* expression could be due to presynaptic and/or postsynaptic changes. If postsynaptic changes underlie the observed phenotype, one would expect corresponding changes in mEPSC amplitude and/or slowing of mEPSC decay kinetics. However, we did not notice corresponding changes in the amplitude, kinetics or charge of mEPSCs in *DTS* mutants (Fig. 4d, light gray and light blue data). On the contrary, we even observed a significant acceleration of mEPSC decay kinetics in *DTS* mutants compared to controls (Fig. 4d, middle). These data suggest that presynaptic mechanisms underlie the proteasome perturbation-induced potentiation of EPSC charge.

**Proteasome function and PHP modulate EGTA-sensitive vesicles**. Presynaptic proteasome interference induced a concomitant increase in EPSC amplitude and a slowing of EPSC decay kinetics, which we attribute to presynaptic changes (Fig. 4). Based on previous work[40,41], this phenotype may be due to additional release of vesicles that are more 'loosely coupled', i.e.,

which are physically more distant from Ca[2+] channels and/or have a lower intrinsic Ca[2+] sensitivity. To test this hypothesis, we probed the effects of the Ca[2+] chelator EGTA on release (Fig. 5a, b). Because of its relatively slow Ca[2+]-binding rate[42], this buffer predominantly interferes with the exocytosis of 'loosely-coupled' vesicles with a lower $p_r$. Application of EGTA-AM (50 μM for 10 min, see Methods) did not significantly change EPSC amplitude or EPSC charge at control synapses, whereas it significantly reduced EPSC charge at *DTS*[pre] mutant synapses (Fig. 5a, b). By contrast, EGTA treatment had no significant effects on mEPSP amplitude (Supplementary Fig. 5a), suggesting that presynaptic proteasome perturbation predominately enhances the release of EGTA-sensitive vesicles.

Based on this result and on the observation that presynaptic proteasome inhibition disrupts PHP (Figs. 1–4), we hypothesized that homeostatic signaling at wild-type synapses may also augment release of EGTA-sensitive vesicles. We therefore next assessed the EGTA sensitivity of release after inducing PHP through PhTX application at wild-type synapses (Fig. 5a, b). Strikingly, EGTA treatment after PhTX application completely blocked the expression of PHP, leading to a pronounced reduction in EPSC amplitude and EPSC charge with respect to synapses that were treated with PhTX alone (Fig. 5a, b). EGTA application had a similar effect on readily releasable vesicle pool (RRP) size assessed by cumulative EPSC amplitude analysis[15,43,44] (Supplementary Fig. 5e). Whereas PhTX-induced glutamate receptor impairment normally results in an increase in RRP size at control synapses[16,44] (Supplementary Fig. 5; see below), we observed reduced cumulative EPSC amplitudes at synapses treated with EGTA and PhTX (Supplementary Fig. 5e).

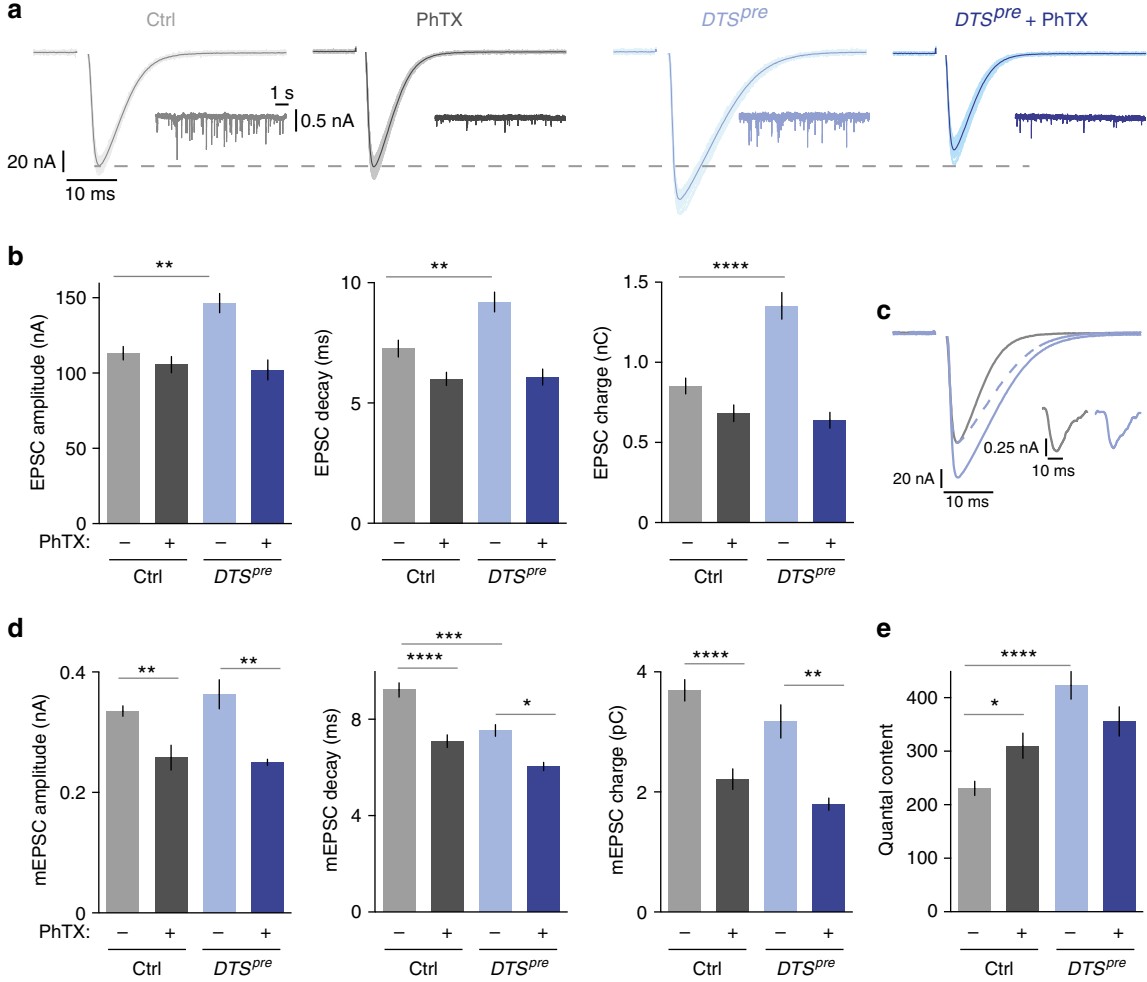

**Fig. 4** Presynaptic proteasome perturbation increases EPSC amplitude and decay time. **a** Two-electrode voltage-clamp recordings of $DTS^{pre}$ mutants and controls with or without PhTX application (20–30 μM; 1 mM $[Ca^{2+}]_e$). Representative EPSCs (50 sweeps per cell; the average is displayed in darker color; stimulation artifacts have been blanked for clarity) and mEPSCs are shown. **b** Quantification of the experiment shown in (**a**). Note that presynaptic proteasome inhibition leads to an increase in EPSC amplitude, decay time constant and charge. Moreover, proteasome inhibition blocks homeostatic plasticity in response to PhTX application. Mean ± s.e.m.; $n \geq 10$ cells; **$p < 0.001$; ****$p < 0.00001$; ANOVA and Tukey's multiple comparison tests. **c** Overlay of an average EPSC of a control cell (gray) and after proteasome inhibition (blue; $DTS^{pre}$). The dashed line shows the EPSC for $DTS^{pre}$ normalized to the EPSC amplitude of control. Inset: representative mEPSCs for control and $DTS^{pre}$. **d** Quantification of mEPSC amplitude, decay and charge. Due to technical reasons, mEPSCs were recorded in different cells than EPSCs. Note that the EPSC decay is faster than the mEPSC decay, likely due to partial glutamate receptor desensitization/saturation. Mean ± s.e.m.; $n \geq 6$ cells; *$p < 0.05$, **$p < 0.001$; ***$p < 0.0001$; ****$p < 0.00001$; ANOVA and Tukey's multiple comparison tests. **e** Quantal content was calculated by the ratio of EPSC charge/mEPSC charge. Mean ± s.e.m.; $n \geq 6$ cells; *$p < 0.05$; ****$p < 0.00001$; ANOVA and Tukey's multiple comparison tests

Thus, the expression of PHP requires additional release of EGTA-sensitive, low-$p_r$ vesicles.

We next explored the relationship between EGTA sensitivity of release, PHP and proteasome function. As shown above, control synapses are significantly less EGTA sensitive under baseline conditions than during PHP (Fig. 5a, b). $DTS^{pre}$-mutant synapses are more EGTA sensitive under baseline conditions than controls (Fig. 5a, b). We next assessed the EGTA sensitivity of release in $DTS^{pre}$ mutants after PhTX application and found that EGTA-AM application led to a decrease in EPSC amplitude and EPSC charge with respect to $DTS^{pre}$ mutants treated with PhTX alone (Fig. 5a, b; two blue bars, right). However, this decrease was significantly less pronounced than the EGTA-induced reduction in EPSC amplitude of PhTX-treated control synapses, indicating that control synapses are more EGTA-sensitive during homeostatic plasticity than $DTS^{pre}$-mutant synapses. Interestingly, the relative decrease in EPSC amplitude induced by EGTA

application after PHP induction was similar between controls and $DTS^{pre}$ mutants when compared to the respective baseline conditions without PhTX or EGTA treatment (ctrl: $-49 \pm 4\%$; $DTS^{pre}$: $-42 \pm 5\%$; Fig. 5a: red bars; Fig. 5b: compare first and last column of each genotype). This demonstrates that the summed EGTA sensitivity of $DTS^{pre}$ mutants under baseline conditions and after homeostatic challenge is similar to the EGTA sensitivity of wild-type synapses during PHP. Two conclusions can be drawn. First, these observations are consistent with the idea that proteasome function and PHP converge on EGTA-sensitive vesicles. Second, our data suggest that homeostatic signaling regulates EGTA-sensitive vesicles more potently than proteasome function.

**Proteasome perturbation increases Ca²⁺ influx and RRP size**. It has been shown that PHP requires an increase in presynaptic

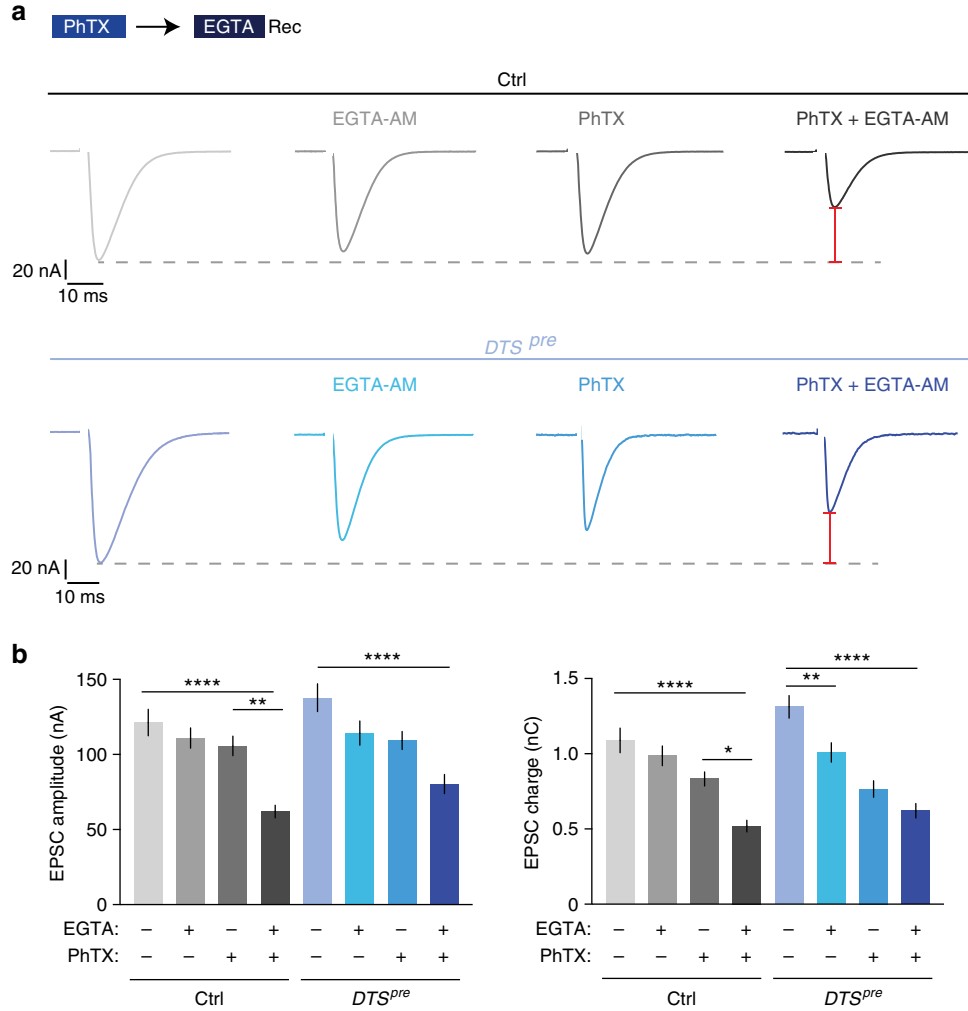

**Fig. 5** Glutamate receptor perturbation and proteasome perturbation enhance release of EGTA-sensitive vesicles. **a** Representative EPSC traces of control (*elav*[C155],'ctrl') and *DTS*[pre] (*elav*[C155]-*Gal4;UAS-DTS5*; '*DTS*[pre]') synapses. The two left EPSC traces from each genotype were recorded from synapses treated with 50 μM EGTA-AM for 10 min or DMSO as control. The last two EPSC traces were recorded from synapses treated with 30 μM PhTX for 10 min, followed by wash and treatment with 50 μM EGTA-AM or DMSO for 10 min (1 mM $[Ca^{2+}]_e$). The red bars symbolize that the decrease after PhTX and EGTA application compared to untreated cells is similar in both genotypes. **b** Quantification of EPSC amplitude and EPSC charge of the experiment shown in (**a**). EGTA-AM application induces a significant decrease in EPSC charge in *DTS*[pre] mutants, whereas there is no significant decrease in EPSC charge at control synapses. After PhTX and EGTA treatment, EPSC amplitudes are significantly smaller in controls than in *DTS*[pre] mutants ($p = 0.0197$; unpaired $t$-test). By contrast, there is no significant difference between both groups in the absence of EGTA treatment ($p = 0.7$), indicating that controls are more EGTA sensitive during homeostatic plasticity than *DTS*[pre] mutants. The increased EGTA-AM sensitivity after PhTX incubation implies that the induction of PHP requires increased recruitment of EGTA-sensitive, low-$p_r$ vesicles. Mean ± s.e.m.; $n \geq 10$ cells; *$p < 0.05$; **$p < 0.001$; ****$p < 0.00001$; ANOVA and Tukey's multiple comparison tests

$Ca^{2+}$ influx[45] and RRP size[16,44]. We therefore investigated if release potentiation upon proteasome perturbation involves similar mechanisms. First, we recorded spatially averaged $Ca^{2+}$ transients at presynaptic boutons in response to single-AP stimulation using two-photon imaging (Fig. 6a–d; terminals were co-loaded with OGB-1 and Alexa-568; see Methods). As shown in Fig. 6b, c, *DTS*[pre] synapses had significantly increased peak amplitudes of presynaptic $Ca^{2+}$ transients with respect to controls. The magnitude of the increase in presynaptic $Ca^{2+}$ influx after proteasome inhibition is relatively large as compared to the effect on neurotransmitter release (Fig. 4a). This observation is in agreement with our electrophysiology data showing that proteasome impairment predominantly potentiates release of EGTA-sensitive, low-$p_r$ vesicles in addition to high-$p_r$ vesicles (Fig. 5). In *DTS*[pre] mutants, we also detected a significant increase in baseline OGB-1-fluorescence intensity before stimulus onset without

corresponding changes in Alexa-568-fluorescence intensity (Fig. 6d, left), suggesting that presynaptic proteasome impairment elevates baseline $Ca^{2+}$ levels. $Ca^{2+}$-transient decay kinetics were similar between mutant and control synapses, indicating that *DTS* expression does not alter $Ca^{2+}$ extrusion and/or $Ca^{2+}$ buffering (Fig. 6d, right). Together, these data demonstrate that presynaptic proteasome perturbation results in increased presynaptic $Ca^{2+}$ influx upon AP stimulation, as well as elevated $Ca^{2+}$ levels at rest. This indicates that presynaptic proteasome function negatively regulates presynaptic $Ca^{2+}$ signaling under baseline conditions.

Next, we investigated RRP size using cumulative EPSC analysis during brief (1 s) 60 Hz trains[15,43,44]. Due to the observation that *DTS*[pre] expression potentiates EPSC charge (Fig. 4), we based our RRP estimate on cumulative EPSC charge. As shown in Fig. 6e, f, RRP analysis of *DTS*[pre] mutants revealed a significant increase in

apparent RRP size with respect to controls. By contrast, there was no significant change in short-term depression during 60-Hz stimulation between $DTS^{pre}$ mutants and controls (Supplementary Fig. 5b), suggesting that the increase in RRP size is unlikely to be mainly due to enhanced presynaptic $Ca^{2+}$ influx. The conclusion that $DTS^{pre}$ also affects release downstream of presynaptic $Ca^{2+}$ influx is further supported by the observation that an increase in $[Ca^{2+}]_e$ from 1 mM to 2 mM does not result in

a slowing of EPSC decay kinetics at wild-type synapses (Supplementary Fig. 5c), in contrast to $DTS^{pre}$ mutants (Fig. 4b). We conclude that the presynaptic proteasome opposes release of low-$p_r$ vesicles through similar mechanisms employed during PHP: the modulation of presynaptic $Ca^{2+}$ influx and RRP size.

Our data imply that presynaptic proteasome impairment predominately recruits vesicles with a low $p_r$ (Fig. 4) that are EGTA sensitive (Fig. 5). To provide further support for this

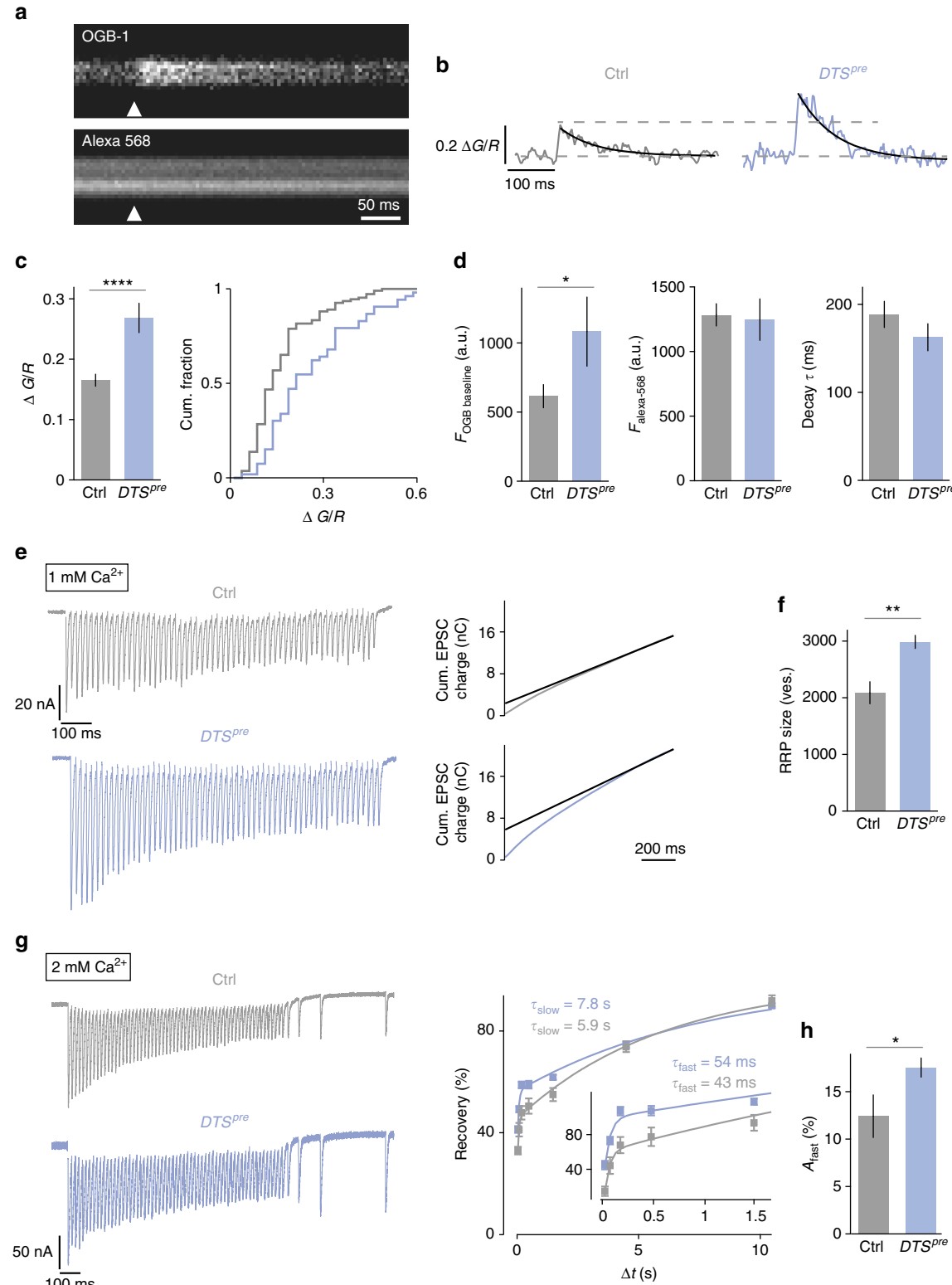

model, we assayed vesicle resupply after RRP depletion induced by high-frequency stimulation at elevated $Ca^{2+}$ concentrations (2 mM $[Ca^{2+}]_e$; Fig. 6g). At many synapses, there are two distinct kinetic components of vesicle replenishment following high-frequency stimulation: a fast recovery time component that likely corresponds to resupply of vesicles with a relatively low $p_r$, and a slow recovery period that is thought to reflect replenishment of vesicles with a higher $p_r$[46–48]. We therefore hypothesized that proteasome perturbation may predominantly enhance the fast recovery component. $DTS^{pre}$ mutants displayed a significantly increased amplitude of the fast recovery component as compared to controls (Fig. 6g, h). There were no significant changes in the fast and slow recovery time constants, as well as the amplitude of the slow recovery component between $DTS^{pre}$ mutants and controls. These data provide indirect evidence that proteasome inhibition results in increased release of low-$p_r$ vesicles that recover rapidly.

**dysbindin and rim are required for release potentiation.** Proteasome perturbation blocks PHP and enhances $Ca^{2+}$ influx and neurotransmitter output. To genetically link proteasome function-dependent control of release and homeostatic plasticity, we assayed synaptic transmission before and after lactacystin application in several PHP mutants (Fig. 7[49]). For quantification, we normalized the mean EPSP amplitude after lactacystin incubation (100 μM for 15 min) to the EPSP amplitude after incubation with saline for 15 min (for absolute data see Table S1). As shown in Fig. 7a, b, most PHP mutants displayed an increase in EPSP amplitude after proteasome interference that was similar to wild-type control synapses (black/gray data). Interestingly, two mutants, rab-3-interacting-protein (rim) and dysbindin (dysb), did not show any change in either EPSP amplitude or quantal content after lactacystin application (red data; Fig. 7a–c). This indicates that these two homeostatic plasticity genes are required for the increase in release upon proteasome inhibition, and therefore provide a link between homeostatic and proteasome-dependent control of release. By extension, based on our observation that proteasome perturbation failed to enhance release in specific PHP mutants, we suggest that the presynaptic proteasome regulates specific presynaptic proteins that are required for homeostatic plasticity.

**Genetic interaction between dysbindin and DTS during PHP.** In a next set of experiments we investigated the role of dysbindin in proteasome-dependent control of release in more detail (Figs. 7d, e, and 8). We focused on dysbindin because dysbindin mutants do not have major defects in baseline synaptic transmission[11], and because the human dysbindin homolog (DTNBP1)

has been linked to schizophrenia[50]. First, we aimed at providing a genetic link between dysbindin and proteasome function in the context of homeostatic plasticity. PHP is completely blocked in homozygous dysbindin mutants[11] (Fig. 8h, i) or after presynaptic expression of two DTS5 copies (Fig. 4). Thus, analysis of homozygous double mutants would not be informative. We therefore tested PhTX-induced PHP in heterozygous double mutants lacking one dysbindin copy and expressing one DTS copy presynaptically ($elav^{c155}$-Gal-4/Y;DTS5/+,dysb/+; Fig. 7d, e). A prerequisite for this analysis is that the heterozygous mutants alone do not impair PHP. PHP proceeds normally in heterozygous dybindin mutants[11]. Similarly, PhTX application induced a homeostatic increase in quantal content that restored EPSP amplitudes to baseline levels at synapses expressing one DTS5 copy (Fig. 7d, e). However, when examining DTS5/+,dysb/+ double heterozygous animals, we observed significantly smaller EPSP amplitudes and no significant increase in quantal content after PhTX incubation (Fig. 7d, e). This finding demonstrates a genetic interaction between dysbindin and DTS during homeostatic plasticity. We also noted a significant increase in EPSP amplitude in DTS5/+,dysb/+ double heterozygous mutants compared to DTS5/+ heterozygous mutants in the absence of PhTX. This suggests that mild proteasome impairment can potentiate release at synapses with decreased dysbindin expression under baseline conditions, but not during homeostatic plasticity. Together, our results demonstrate a genetic link between dysbindin and proteasome function during PHP and baseline synaptic transmission.

**Dysbindin controls release of EGTA-sensitive vesicles.** Acute proteasome perturbation failed to potentiate release at homozygous dysbindin mutant synapses (Fig. 7). We therefore hypothesized that Dysbindin protein levels may be important for the regulation of neurotransmitter release. To test this hypothesis, we first assayed release after presynaptic overexpression of wild-type venus-dysbindin (Fig. 8a–g). A similar construct was previously shown to rescue the dysbindin mutant phenotype and to potentiate release[11]. Consistent with previous data[11], venus-tagged Dysbindin localized to presynaptic NMJ boutons (Fig. 8a). We detected a significant increase in EPSC amplitude and EPSC charge upon single-AP stimulation (Fig. 8b, c), as well as a significant increase in cumulative EPSC amplitude during 60 Hz trains upon presynaptic dysbindin overexpression (Fig. 8d, e), indicating a role for dysbindin expression levels in the regulation of RRP size.

We provided evidence that both genetic proteasome perturbation and PHP induction result in increased recruitment of EGTA-sensitive vesicles (Fig. 5). We next asked if the increase in release seen after dysbindin overexpression is also due to enhanced

**Fig. 6** Proteasome perturbation increases presynaptic $Ca^{2+}$ influx, RRP size and the amplitude of the fast recovery phase. **a** Representative two-photon line scans of a control bouton after loading with OGB-1 and Alexa-568. Arrowhead indicates AP-stimulus onset. **b** Representative spatially averaged $Ca^{2+}$ transients of a control synapse and a $DTS^{pre}$ synapse (average of 10 sweeps). **c** Quantification of $Ca^{2+}$-transient peak amplitudes of control and $DTS^{pre}$ synapses and cumulative frequency plot. Proteasome inhibition results in a significant increase in the peak amplitude compared to control cells. **d** Baseline fluorescence of the $Ca^{2+}$ indicator OGB-1 and the $Ca^{2+}$-independent dye Alexa-568 in the two genotypes. Baseline OGB-1 fluorescence is significantly higher in $DTS^{pre}$ synapses. The decay time constant $\tau$ is not changed. Mean ± s.e.m.; $n \geq 53$ boutons; *$p < 0.05$; ****$p < 0.00001$; Wilcoxon rank-sum test. **e** Left: Representative EPSC trains in response to 60-Hz stimulation (60 stimuli, 1 mM $[Ca^{2+}]_e$) for control cells and $DTS^{pre}$ mutant cells. Right: cumulative EPSC charge was calculated by back-extrapolation of a linear fit (black line) to the last 15 stimuli of the cumulative EPSC integrals to time point zero. **f** RRP size was calculated by dividing cumulative EPSC charge by the average mEPSC charge. **g** Left: Example of EPSCs during 60-Hz stimulation (60 stimuli, 2 mM $[Ca^{2+}]_e$) followed by recovery pulses given at intervals of 25, 75, 175 and 475 ms after the last train stimulus of the indicated genotypes. Right: Average recovery (mean EPSC amplitude at a given interval divided by the first EPSC amplitude of the 60 Hz train) versus recovery interval ($\Delta t$) of the indicated genotypes superimposed with biexponential fits. Inset shows the data at short intervals. Biexponential fits of average data gave the time constants noted in the figure. **h** Average amplitude of the fast recovery component '$A_{fast}$' of the two groups obtained from bi-exponential fits of individual synapses. Note the significant increase in the fast recovery component in $DTS^{pre}$ mutants (there was no difference in $A_{slow}$ and recovery kinetics between the two groups). Mean ± s.e.m.; $n \geq 10$ cells; *$p < 0.05$; **$p < 0.001$; Student's t-test

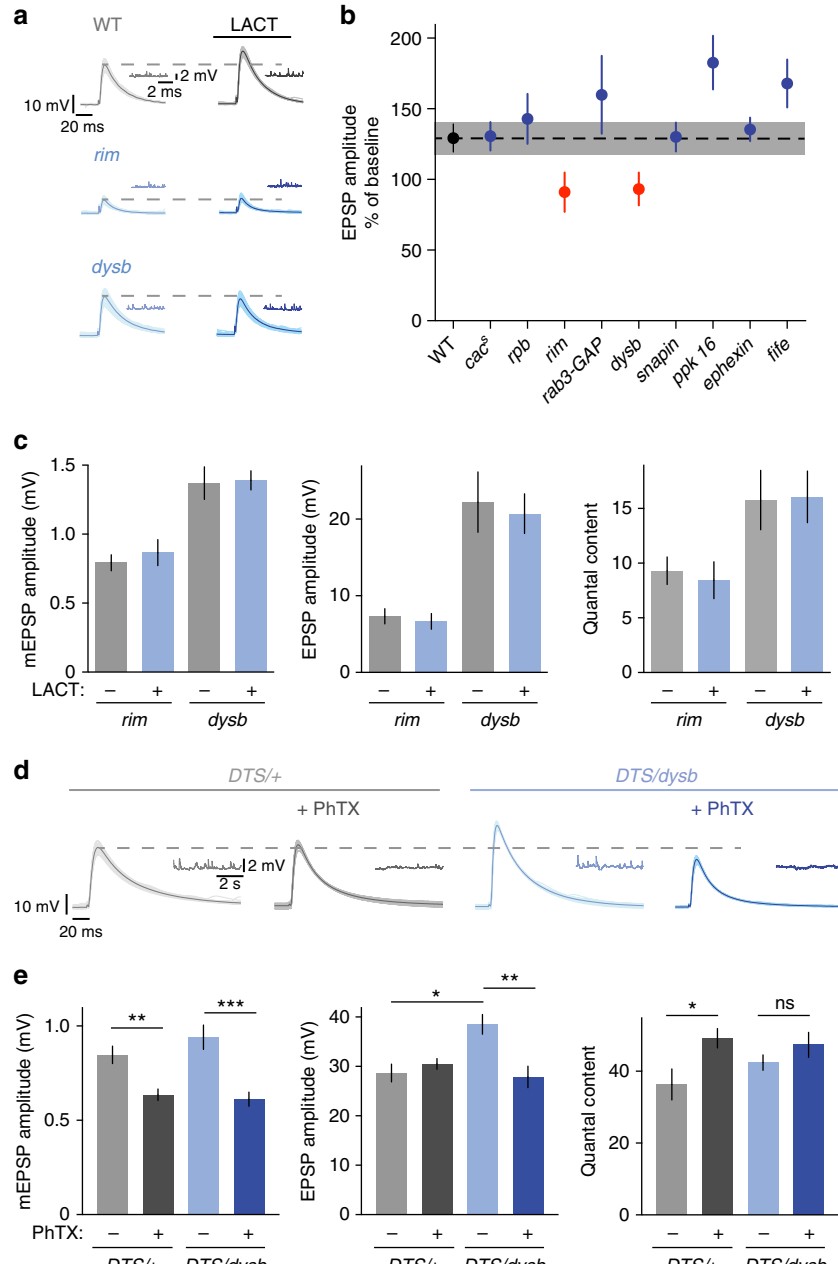

**Fig. 7** *rim* and *dysbindin* are required for proteasome-dependent potentiation. **a** Application of 100 μM lactacystin for 15 min to wild-type synapses and selected PHP mutants. Representative EPSP and mEPSP traces are shown for wild type, *rim* and *dysbindin* mutants. **b** For quantification, EPSP amplitude after lactacystin application was normalized to the EPSP amplitude at baseline. Wild-type EPSPs show an increase to 130% upon lactacystin treatment. The PHP mutants *cac^s*, *rim-binding protein* (*rbp*), *rab3-GAP*, *snapin* (presynaptic RNAi), *pickpocket16* (*ppk 16*), *ephexin* and *fife* show similar or stronger increases, whereas *rim* and *dysbindin* (*dysb*) mutants do not respond to lactacystin treatment. Due to defects in baseline EPSP amplitude in some of the genotypes, different $[Ca^{2+}]_e$ concentrations were used to adjust baseline EPSP amplitudes to comparable values: WT: 0.3 mM; *cac^s*: 0.5 mM; *rbp*: 1 mM; *rim*: 0.3 mM; *rab3-GAP*: 0.3 mM; *dysbindin*: 0.3 mM; *snapin RNAi*: 0.3 mM; *pickpocket 16*: 0.25 mM; *ephexin*: 0.25 mM; *fife*: 0.3 mM. Mean ± s.e.m.; $n \geq 8$ cells. **c** Quantification of mEPSP amplitude, EPSP amplitude and quantal content of *rim* and *dysbindin* mutant synapses at baseline and after lactacystin application (100 μM for 15 min). Mean ± s.e.m.; $n \geq 10$ cells. **d** Synapses expressing one allele of the mutant proteasome subunit presynaptically either in a wild-type background (*DTS/+*; het) or in larvae mutant for one allele of *dysbindin* (*DTS/dysb*; double het) were measured under baseline conditions and after application of 20 μM PhTX for 10 min (0.3 mM $[Ca^{2+}]_e$). Representative EPSP and mEPSP traces are shown. **e** Quantification of mEPSP amplitude, EPSP amplitude and quantal content for the experiment described in (**d**). *DTS/+*heterozygous synapses have no defect in PHP, whereas *DTS/dysb* double heterozygous synapses show no increase in quantal content after PhTX application. *$p < 0.05$, **$p < 0.001$; ***$p < 0.0001$

recruitment of EGTA-sensitive vesicles. Application of EGTA-AM to synapses overexpressing venus-*dysbindin* resulted in a significant decrease in EPSC amplitude (Fig. 8f, g), similar to the decrease seen after EGTA-AM application to *DTS^pre* synapses (Fig. 5). EGTA-AM treatment of *dysbindin*-overexpressing synapses also significantly reduced cumulative EPSC amplitudes compared to untreated NMJs overexpressing *dysbindin* (Supplementary Fig. 5e). These results suggest that *dysbindin* overexpression potentiates release from an EGTA-sensitive vesicle pool.

Wild-type synapses display a high EGTA sensitivity upon PhTX treatment (Fig. 5). If *dysbindin* is involved in homeostatic regulation of EGTA-sensitive vesicles, we expect release to be less EGTA sensitive after PhTX application in *dysbindin* mutants. To test this hypothesis, we recorded EPSCs in *dysbindin* mutants after incubation with PhTX and PhTX+EGTA-AM (Fig. 8h, i).

First, we observed that baseline synaptic transmission was not changed in *dysbindin* mutants compared to wild-type controls (Fig. 8h, i, left; 1 mM $[Ca^{2+}]_e$), in line with a previous study[11]. Application of PhTX led to a significant decrease in EPSC amplitude and EPSC charge, indicating that PHP is impaired in *dysbindin* mutants, consistent with earlier work[11]. However, and

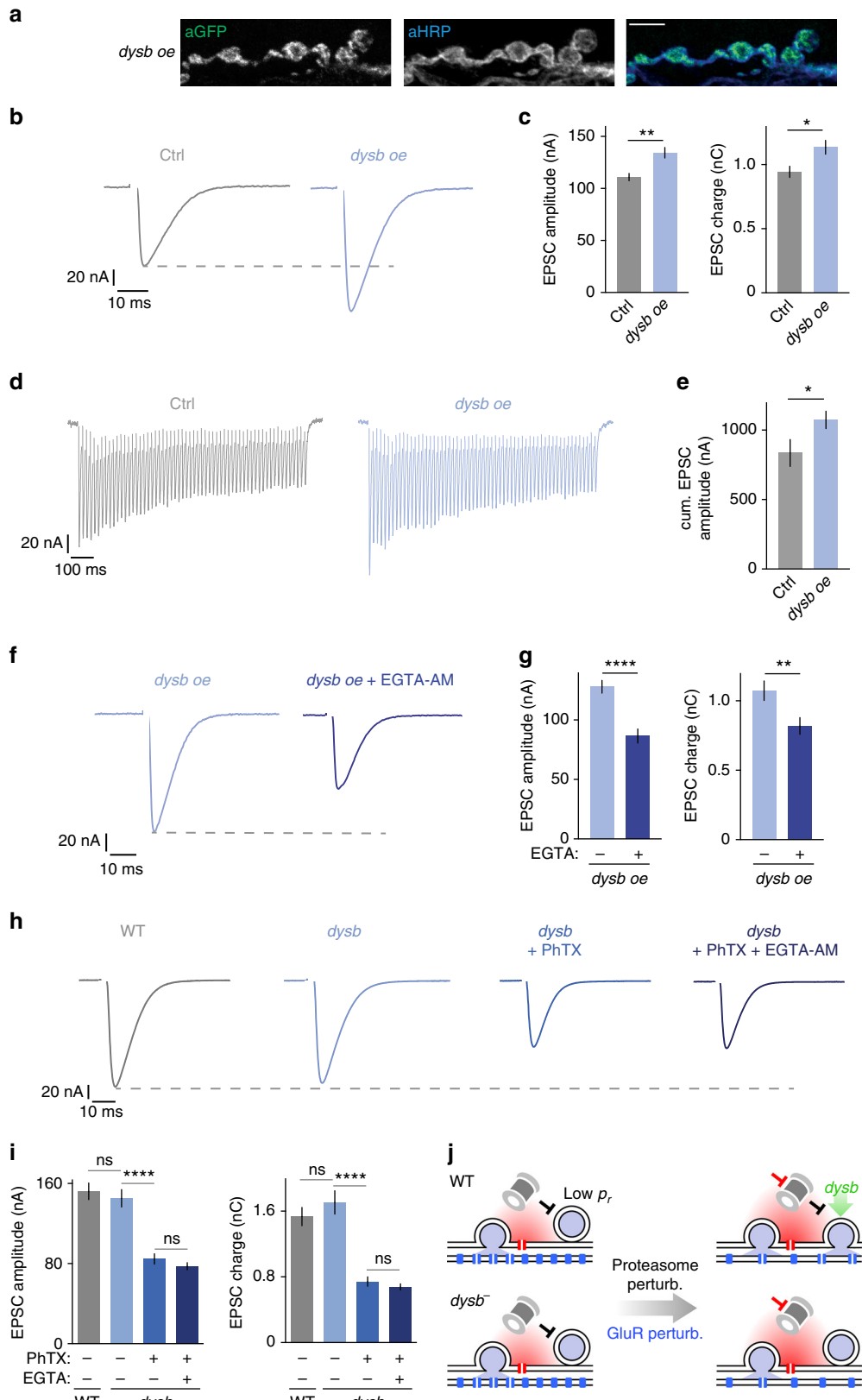

in contrast to wild-type synapses, application of EGTA-AM onto PhTX-treated *dysbindin* mutants did not result in significant changes in EPSC amplitude or charge (Fig. 8h, i). These data support the conclusion that the enhanced release of EGTA-sensitive vesicles during PHP requires *dysbindin*.

## Discussion

Here we demonstrate that presynaptic proteasome function is indispensable for PHP induced by glutamate receptor perturbation at the *Drosophila* NMJ. Moreover, we provide evidence that an EGTA-sensitive pool of synaptic vesicles is regulated by proteasome function, homeostatic plasticity and *dysbindin*.

We observed rapid effects of proteasome perturbation on neurotransmitter release and the abundance of ubiquitinated proteins on the minute time scale. These data suggest that the proteasomal degradation rate is relatively high under baseline conditions, consistent with previous work on local protein degradation in the presynaptic or postsynaptic compartment[2,3], and considerably faster than the average neuron-wide turnover rates of synaptic proteins (2–5 days[1]). Such rapid degradation rates could in turn allow for potent modulation of protein abundance and/or ubiquitination through regulation of UPS function during PHP.

Perturbation of proteasome function has diverse effects on cellular physiology, such as altering the levels of free ubiquitin and mono-ubiquitinated proteins, activating macroautophagy or upregulation of lysosomal enzyme levels[51–53]. The observed phenotypes could thus be due to indirect effects of impaired proteasome function on synaptic physiology. Even if we cannot rule out this possibility, several lines of evidence argue against a major contribution of indirect effects. First, acute or prolonged proteasome perturbation does not affect release at synapses that express PHP (Fig. 2). Second, we provide genetic evidence that proteasome perturbation-induced changes in release are blocked in two PHP mutants (Fig. 7). Third, we detect no major changes in synaptic morphology or synaptic development upon prolonged proteasome perturbation (Fig. 3). Fourth, interfering with proteasome function does not impair neurotransmitter release, but rather results in a net increase in release (Figs. 1, 2, 4, 5, 6 and 7). Taken together, these data suggest that proteasome function opposes release by degrading proteins under baseline conditions, and that normal degradation of these proteins is required for PHP.

Our genetic data imply that not all proteins required for PHP are regulated through UPS-dependent degradation under our experimental conditions (pharmacological proteasome perturbation for 15 min), because release can be potentiated upon brief proteasome inhibition in most PHP mutants (Fig. 7). This observation is consistent with a recent study demonstrating that the abundance of most synaptic proteins does not change after prolonged pharmacological proteasome perturbation in cultured mouse hippocampal neurons[52]. Based on our observation that PHP is blocked in *cut-up* mutants (Supplementary Fig. 2), which were recently shown to have a defect in proteasome trafficking[36], it is conceivable that proteasome mobility and/or recruitment are modulated during PHP.

We provide evidence that proteasome perturbation and homeostatic signaling recruit EGTA-sensitive vesicles with a lower $p_r$ in addition to vesicles with higher $p_r$. Previous work revealed that tightly-coupled, high-$p_r$ vesicles are required for PHP[16]. PHP therefore likely involves vesicle pools with different $p_r$. Homeostatic regulation of EGTA-sensitive, loosely-coupled vesicles depends on *dysbindin* (this study), whereas tightly-coupled vesicles are controlled by RIM-binding protein[16]. Together, these results imply that PHP involves two genetically separable populations of vesicles with different $p_r$.

Vesicle pools with different $p_r$ and release kinetics have been observed at various synapses[40,46–48,54], and these pools might be differentially regulated during synaptic plasticity. Here we provide evidence that *dysbindin* is required for the recruitment of EGTA-sensitive vesicles during PHP. It will be exciting to investigate the roles of other presynaptic proteins that have been implicated in the release of EGTA-sensitive vesicles, such as Tomosyn[54], in the context of proteasome degradation and PHP. Interestingly, a recent study at the mouse NMJ observed accelerated release kinetics of 'slow' synaptic vesicles during PHP[55]. Thus, homeostatic potentiation of vesicles with lower $p_r$/slower release kinetics may be an evolutionarily conserved mechanism.

Which mechanisms could potentiate the release of low-$p_r$ vesicles? We uncovered that presynaptic proteasome perturbation results in enhanced presynaptic $Ca^{2+}$ influx, independent of major changes in $Ca^{2+}$ buffering and/or extrusion. We therefore consider changes in presynaptic $Ca^{2+}$ influx as a possible mechanism underlying the increase in low-$p_r$ vesicle release. Proteasome perturbation also increased RRP size. Earlier work revealed that changes in presynaptic $Ca^{2+}$ influx modulate release in part by altering apparent RRP size[16,43,56]. The increase in RRP size or EGTA sensitivity upon proteasome inhibition may therefore be in part a secondary consequence of enhanced presynaptic $Ca^{2+}$ influx. However, several observations, such as the increased amplitude of the fast recovery phase after pool depletion or the slowing of EPSC decay kinetics upon presynaptic

**Fig. 8** Dysbindin overexpression increases release of EGTA-sensitive vesicles that are required for PHP. **a** Immunostaining of a larval NMJ (muscle 6/7) overexpressing *venus-dysbindin* presynaptically. Venus-Dysbindin was detected with anti-GFP (aGFP), neuronal membrane with an antibody against horseradish peroxidase (aHRP); scale bar = 5 μm. **b** Representative EPSCs of controls (*elav*[C155]) and after presynaptic *venus-dysbindin* overexpression (*elav*[C155]-Gal4/Y;;UAS-venus-dysbindin/+; 'dysb oe') at 1 mM $[Ca^{2+}]_e$. **c** Quantification of EPSC amplitude/charge of the experiment shown in (**b**). *dysbindin* overexpression increases EPSC amplitude and charge compared to controls. Mean ± s.e.m.; $n ≥ 14$ cells; *$p < 0.05$; **$p < 0.001$; Student's $t$-test. **d** Representative EPSC trains in response to 60-Hz stimulation (60 stimuli, 1 mM $[Ca^{2+}]_e$) for controls and *venus-dysbindin* overexpressing NMJs. **e** Cumulative EPSC charge was calculated by back-extrapolation of a linear fit (black line) to the last 15 stimuli of the cumulative EPSC integrals to time point zero. Mean ± s.e.m.; $n ≥ 10$ cells; *$p < 0.05$; Student's $t$-test. **f** Representative EPSCs after presynaptic *venus-dysbindin* overexpression and application of 50 μM EGTA-AM or DMSO for 10 min (1 mM $[Ca^{2+}]_e$). **g** Quantification of EPSC amplitude/charge of the experiment shown in (**f**). Application of EGTA-AM significantly decreased EPSC amplitude/charge, suggesting that *dysbindin* overexpression enhances the recruitment of EGTA-sensitive vesicles. Mean ± s.e.m.; $n ≥ 18$ cells; **$p < 0.001$; ****$p < 0.00001$; Student's $t$-test. **h** Representative EPSCs of wild-type (*w*[1118]) and *dysbindin* mutants under baseline conditions and after treatment with PhTX or PhTX+EGTA-AM (1 mM $[Ca^{2+}]_e$). **i** Quantification of EPSC amplitude/charge of the experiment shown in (**h**). EPSC amplitudes of *dysbindin* mutants are similar to wild-type under baseline conditions, but decreased after PhTX application, indicating a defect in PHP. Additional treatment with EGTA-AM does not further reduce EPSC amplitude/charge. **j** Summary and model: Top: Glutamate receptor perturbation or proteasome perturbation induce an increase in presynaptic $Ca^{2+}$ influx[40] and enhance release of EGTA-sensitive, low-$p_r$ vesicles. Proteasome perturbation blocks PHP. Our data suggest that proteasome function opposes the release of EGTA-sensitive vesicles (black 'T') that are potentiated during homeostatic plasticity. Bottom: *dysbindin* mutants fail to increase release upon glutamate receptor perturbation[11] or proteasome perturbation, implying that *dysbindin* controls the release of EGTA-sensitive vesicles

proteasome perturbation, which are not seen after increasing $[Ca^{2+}]_e$, indicate that the increase in release is not caused by presynaptic $Ca^{2+}$ influx alone. Together, these data imply that a combination of increased presynaptic $Ca^{2+}$ influx and RRP size underlie the enhancement of release after proteasome or glutamate receptor perturbation. Which molecular mechanisms may link UPS function to the modulation of presynaptic $Ca^{2+}$ influx or RRP size? Genetic data suggest that *dysbindin* functions independently of presynaptic $Ca^{2+}$ influx[11]. Interestingly, *rim* mutants were shown to have a defect in homeostatic RRP size modulation, but unchanged homeostatic control of presynaptic $Ca^{2+}$ influx[15]. We therefore speculate that RIM and Dysbindin may be involved in proteasome-dependent RRP size regulation.

How do our observations relate to mammalian synapses? At cultured hippocampal rat synapses, proteasome inhibition augments recycling vesicle pool size[29] or prevents a decrease in RRP size induced by prolonged (4 h) depolarization[25]. Moreover, several studies at mammalian synapses suggest that UPS-dependent regulation of RIM abundance regulates neurotransmitter release during baseline synaptic transmission and synaptic plasticity[25,26,30]. Finally, there is evidence that ubiquitination acutely regulates release on the minute time scale at cultured hippocampal rat synapses[27]. Together, these studies suggest that rapid, UPS-dependent control of RRP size may be evolutionarily conserved. Future studies will further elucidate the molecular signaling pathways relating UPS function to neurotransmitter release during baseline synaptic transmission and homeostatic plasticity.

## Methods

**Fly stocks and genetics**. *Drosophila* stocks were maintained at 21 °C–25 °C on normal food. The $w^{1118}$ strain was used as a wild-type control. The $GluRIIA^{SP16}$ null mutation[10], *rim103*[15], *rbpSTOP1*, *rbpS2.01*[57], *rab3-GAP*[14], *dysbindin*[11], *snapin* RNAi[21], *cac*[5], *exn*$^{EY10953}$[13] and *ppk16*$^{166}$[18] were a kind gift from the lab of Graeme Davis. *Cut-up*[36] (*ctp*) mutant flies were a kind gift from Benjamin Eaton, and *fife*[58] mutants were kindly provided by Kate O'Connor-Giles. For pan-neuronal expression, the *elav*$^{C155}$-*Gal4* (on the X-chromosome) driver line was used and male larvae were recorded. For expression in muscle cells, the *24B-Gal4* driver line was used. Both driver lines, as well as the *UAS-DTS5* (*Prosβ6*[35]) and the *UAS-DTS5;DTS7* (*Prosβ6*; *Prosβ2*[35]) lines, were obtained from Bloomington *Drosophila* Stock Center (Bloomington, IN, USA). One or two *DTS* copies were overexpressed at a permissive temperature of 25 °C throughout development[35]. To obtain *elav*; $GluRIIA^{SP16}$, *DTS5* flies, $GluRIIA^{SP16}$ mutant flies were recombined on the second chromosome with *UAS-DTS5* flies and crossed with *elav*$^{C155}$-*Gal4* for a stable expression of *UAS-DTS5* in the $GluRIIA^{SP16}$ mutant background. Standard second and third chromosome balancer lines (Bloomington) and genetic strategies were used for all crosses and for maintaining mutant lines. For the generation of the transgenic *venus-dysbindin* overexpressing flies (Fig. 8), constructs based on the *pUAST-attB-FLAG* vector backbone were injected into the *ZP-attP-86Fb* fly line harboring a landing site on the third chromosome according to standard procedures[59].

**Electrophysiology**. Electrophysiological recordings were made from third-instar larvae at the wandering stage. Larvae were dissected and sharp-electrode recordings were made from muscle 6 in abdominal segments 2 and 3 using an Axoclamp 900A amplifier (Molecular Devices) or an EPC 10/2 amplifier (HEKA)[9]. The extracellular HL3 saline contained (in mM): 70 NaCl, 5 KCl, 10 $MgCl_2$, 10 $NaHCO_3$, 115 sucrose, 5 trehalose, 5 HEPES, 0.3 $CaCl_2$ for single-electrode recordings (Figs. 1 and 2; for Fig. 7 see figure legend) or 1 $CaCl_2$ for TEVC recordings and $Ca^{2+}$ imaging experiments (Figs 4, 5, 6 and 8). To induce PHP, larvae were incubated in 20 μM Philanthotoxin-433 (Sigma-Aldrich, P207) (to induce a reliable effect for TEVC recordings, 30 μM PhTX was used) for 10 min at room temperature. For proteasome inhibition, larvae were incubated in 100 μM lactacystin (Sigma-Aldrich, L6785) for 15–45 min (see figure legends) at room temperature (for Supplementary Fig. 1, concentration and incubation time as indicated) or 50 μM MG-132 (Sigma-Aldrich, SML1135). After PhTX and lactacystin treatment, larvae were washed with HL3, dissected and recordings were carried out.

For EGTA-AM treatment, larvae were dissected before treatment, incubated with 50 μM EGTA-AM (Molecular Probes, E1219) for 10 min at room temperature and washed thoroughly with HL3, before recordings were carried out.

From each muscle cell, 30 AP-evoked EPSPs were recorded (stimulus duration, 1 ms) and averaged. At least 100 mEPSPs were recorded and averaged for each cell to obtain the average mEPSP amplitude. Quantal content was calculated by dividing the average EPSP amplitude by the average mEPSP amplitude for each cell.

TEVC recordings were performed using an Axoclamp 900A Amplifier (Molecular Devices). Cells were clamped to a membrane potential of −60 mV. A total of 50 EPSCs were averaged to obtain the mean EPSC (stimulus duration, 1 ms) value for each cell. AP-evoked EPSCs were recorded with a combination of a HS-9A ×10 and a HS-9A ×1 headstage (Molecular Devices), whereas mEPSC recordings were carried out using two HS-9A ×1 headstages (Axon CNS, Molecular Devices). The size of the RRP of synaptic vesicles was calculated by the method of cumulative EPSC charge[32,43]. Synapses were stimulated with 60 Hz trains (60 stimuli, 5 trains per cell). The cumulative EPSC charge was obtained by back-extrapolating a linear fit to the last 15 cumulative EPSC charge values of the 60 Hz train to time zero. In cases where EGTA-AM had been applied (Supplementary Fig. 5), cumulative EPSC amplitude was calculated with the Elmqvist and Quastel method[60,61]. Here, a linear fit to the first two points of a plot of peak EPSC amplitude versus cumulative EPSC amplitude was extrapolated to the abscissa. This extrapolation gave the cumulative EPSC amplitude for the RRP size estimate. For the recovery experiments, synapses were stimulated with a 60 Hz train for 1 s (2 mM $[Ca^{2+}]_e$) followed by single stimuli applied at 0.025 s, 0.075 s, 0.175 s, 0.475 s, 1.475 s, 4.475 s and 10.475 s after the train. Average EPSC amplitudes during recovery were normalized to the first EPSC amplitude of the 60 Hz train and fitted with a double-exponential constrained to an amplitude of one.

**$Ca^{2+}$ imaging**. Wandering third-instar larvae were dissected in $Ca^{2+}$-free HL3, leaving brain and nerves intact[45]. Nerves were cut close to the NMJ in the presence of $Ca^{2+}$-free HL3 containing 5.6 mM Oregon Green 488 Bapta-1 and 1 mM Alexa-568 (both from Invitrogen). Thereafter, the preparation was put on ice for 5 min and washed thoroughly for another 5 min. After changing the extracellular solution to HL3 containing 1 mM $Ca^{2+}$, spatially averaged $Ca^{2+}$ transients in response to single-AP stimulation (stimulus duration, 3 ms) were imaged in type 1b boutons synapsing onto muscle 6/7 of abdominal segments A2/A3 with a two-photon microscope (Scientifica). The 780 nm excitation light of a Mai Tai laser (Spectra Physics) was focused onto individual boutons with a 40× water-dipping objective (Olympus, 1.0 NA), and line scans across single boutons were made at 333 Hz for 1 s. The average fluorescence change per bouton is based on 10 sweeps (inter-sweep interval, 10 s). Traces were background subtracted and corrected for bleaching using an exponential fit. If the transient increase in OGB-1 fluorescence could not properly be fit with a single exponential, the $\tau$ value was excluded from analysis. Changes in fluorescence were quantified as $\Delta G/R = (G_{peak} - G_{base})/R$, where $G_{peak}$ is the peak fluorescence intensity in response to stimulation, and $G_{base}$ and $R$ are the average fluorescence intensity during 150 ms before stimulation in the green and red channel, respectively.

**Data analysis**. Electrophysiology data were acquired with Clampex (Axon CNS, Molecular Devices) or PatchMaster (HEKA) and analyzed with custom-written routines in Igor Pro (Wavemetrics). Spontaneous mEPSPs and mEPSCs were analyzed with Mini Analysis Software (Synaptosoft). Average mEPSC waveforms were obtained with AxoGraph X. $Ca^{2+}$ imaging data were acquired with Helio-Scan[62] and analyzed with custom-written Igor Pro routines. Statistical analysis was done with Prism software (GraphPad software). Data were tested for normality using D'Agostino–Pearson omnibus normality test. Normally distributed data were analyzed for statistical differences via *t*-test (pairwise comparison) or analysis of variance (ANOVA) and Tukey's test for multiple comparisons. For non-normally distributed data, Wilcoxon rank-sum test or Dunn's multiple comparisons after nonparametric ANOVA were used. Data are given as mean ± s.e.m.

**Immunohistochemistry**. Third-instar larval preparations were fixed for 2 min with Bouin's fixative (100%, Sigma-Aldrich, HT-10132), or 100% ice-cold Ethanol for 5 min. For proteasome blockade, larvae were incubated with 100 μM lactacystin for 15, 30 or 45 min at room temperature before applying the fixative. Control larvae were incubated with HL3 saline for the same time durations. Preparations were washed thoroughly with PBS containing 0.05% Triton X-100. After washing, preparations were blocked with 3% normal goat serum in PBS containing 0.05% Triton X-100. Incubation with the primary antibody was done at 4 °C on a rotating platform overnight. The following primary antibodies and dilutions were used in this study: anti-ubiquitin (FK2, mouse, Calbiochem ST1200; 1:5000), anti-discs large (rabbit, DSHB AB_2617529; 1:10,000), anti-Bruchpilot (mouse, nc82; 1:100), anti-GFP (rabbit, Life Technologies G10362; 1:1000) and anti-HRP Alexa-Fluor 647 (goat, Jackson ImmunoResearch 123–605–021; 1:200).

Secondary antibodies (Life Technologies; 1:500) were applied for 2 h at room temperature on a rotating platform. Preparations were mounted onto slides with ProLong Gold (Life Technologies, P36930). Images were acquired on an LSM710 confocal microscope (Carl Zeiss), using Zen software (Carl Zeiss). Image analysis was carried out in ImageJ. Quantification of fluorescence intensity was done on maximal projections. Figures were assembled using Adobe Photoshop and Illustrator software.

**Data availability**. The data that support the findings of this study are available from the corresponding author upon reasonable request.

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

## Acknowledgements

This research was funded by a 'Forschungskredit' grant of the University of Zurich to C. W. and a Swiss National Science Foundation Assistant Professor grant (PP00P3–15) to M.M. We thank Philipp Bethge and Fritjof Helmchen for sharing the 2P set-up and for providing technical support. Moreover, we thank Marc Debrunner for *Drosophila* embryo injections. We are grateful to Grae Davis, Dion Dickman and members of the Müller lab for helpful discussions and critical comments on the manuscript.

## Author contributions

M.M. and C.W. conceptualized and designed experiments. C.W., I.D., S.S., O.G. and M.M. conducted research, analyzed and interpreted data. C.W. and M.M. wrote the manuscript.

## Additional information

**Competing interests:** The authors declare no competing financial interests.

