## [Peer Review File · Nature Communications]

Reviewers' comments:

Reviewer #1 (Remarks to the Author):

This paper examines presynaptic mechanisms underlying homeostatic plasticity at the *Drosophila* NMJ. This preparation has proven to be a powerful system in which to study homeostatic signaling at synapses, and has been used by several groups to identify genetic and molecular determinants of synaptic compensation. Here, the authors primarily focus on the presynaptic role of the proteasome, and argue that presynaptic protein degradation opposes the release of "loosely coupled," low-release probability vesicles that are recruited during homeostatic regulation of quantal content. They go on to conclude that the gene *Dysbindin*, which has previously been implicated in presynaptic homeostasis at the fly NMJ, plays a key role in regulating access to these "loosely-coupled" vesicles. As such, the authors do a nice job of integrating their novel findings within a mechanistic framework that has been established by the powerful genetics of this system.

Overall, the experiments have been thoughtfully executed with appropriate controls. The findings reveal novel roles of proteasome-dependent degradation and *Dysbindin* in homeostasis and are likely to be of broad interest to those in the field. However, there are several issues that need to be addressed prior to publication:

MAJOR CONCERNS

1) A moderate concern related to the approach is that proteasome disruption, particularly chronic proteasome disruption, is likely to have numerous secondary effects that profoundly alter cell physiology. This can complicate the rather simple interpretation that degradation of some pool of presynaptic proteins is directly tied to vesicle release. The authors are quite up front about this issue, and do a reasonable job of addressing it – for example, using the genetic strategy to complement the pharmacological inhibitors. However, it would be nice to see a little more validation of the main effects using other proteasome inhibitors (e.g., epoximycin, MG132), and these experiments seem fairly straightforward. In addition, the authors should probably weigh the secondary impact of proteasome inhibition a little more strongly in their interpretation of a role for protein degradation specifically, in presynaptic homeostasis. For example, it is well known that strong proteasome inhibition can deplete the free ubiquitin pool in cells, which can impact other cellular events controlled by mono-ubiquitination. Some discussion of this issue is warranted.

2) An important tool used throughout the paper is the genetic inhibition of the proteasome through expression of mutant proteasome subunits. This tool is the basis for the presynaptic role of the proteasome the authors identify, which is a central finding of the paper. Surprisingly, however, the authors provide little in the way of rationale for this approach. On page 7, they describe the basic strategy – overexpression of one or two dominant temperature-sensitive mutant proteasome subunits – but do not articulate why this strategy impairs proteasome function. They refer the reader to the methods, but very little information is provided there either. It is not clear why expression of these mutant subunits impairs proteasome function, and no mention is made of shifting temperatures as is standard with temperature-sensitive mutants. Given the central importance of this tool to the conclusions, the authors must provide more information regarding this approach, and also provide data demonstrating how effectively this strategy achieves the desired effect – i.e., how strongly proteasome function is disrupted in flies expressing these mutant subunits. Much more needs to be done here.

3) The authors define a pool of synaptic vesicles as "loosely-coupled," due to the fact that the slow Ca^{2+} chelator EGTA apparently reduces their release. The logic is that slow chelators do not abolish release because the site of Ca^{2+} entry is so close to the primed vesicles, which are "tightly coupled" to Ca^{2+} influx. By extension, then, the interpretation is that vesicles sensitive to EGTA must be more loosely coupled to the site of Ca^{2+} entry. However, there are numerous Ca^{2+} -dependent events (e.g., priming), aside from direct vesicle fusion, that regulate transmission from presynaptic terminals,

and EGTA could be impacting these. And slow chelators like EGTA have been shown time-and-time again to disrupt presynaptic short-term plasticity such as facilitation, PTP, and augmentation that arise due to Ca^{2+} -dependent changes in release probability. Moreover, the authors demonstrate that basal Ca^{2+} levels are increased with proteasome inhibition, suggesting that the EGTA-dependent effects they observe may simply be due to reducing free Ca^{2+} independent of fusion. Given these issues, it is important for the authors to provide other evidence (aside from EGTA sensitivity) for this presumably distinct vesicle pool.

4) The experiments provide little in the way of mechanistic insight into how Dysbindin recruits the novel population of low Pr vesicles. Dysbindin has already been established as an essential homeostatic plasticity gene, so the fact that it regulates release and EGTA sensitivity is not altogether surprising. Are the levels of Dysbindin altered during homeostatic plasticity, and is this mostly via a change in half life as might be expected for a target regulated by the proteasome? By what mechanism does Dysbindin control the availability of the low Pr vesicle pool? The paper would be stronger if the authors could provide more data what speak directly to these questions.

SPECIFIC COMMENTS

1) The authors discuss (on page 6) how their results on the effects of proteasome inhibition extend previous work from the *Drosophila* NMJ. There is a fairly substantial literature on this from mammalian central synapses as well that should be included here.

2) The authors do a nice job of showing raw values for EPSP and mEPSP amplitude, in addition to the computed changes in quantal content for all of their manipulations. However, they never mention how mEPSP frequency is altered in these experiments? I realize that many papers fail to mention mEPSP frequency, but this is another source of information that is clearly relevant to presynaptic mechanisms. Especially given recent evidence that evoked and spontaneous release may be uniquely regulated, it is worth providing this data.

Reviewer #2 (Remarks to the Author):

This study addresses contributions of the presynaptic proteasome to homeostatic control of synaptic transmission at the *Drosophila* neuromuscular junction. The interrelationship between proteasome-dependent potentiation and homeostatic control is documented in detail (6 figures), followed by experiments to link two presynaptic proteins, RIM and especially Dysbindin, to these two processes. The experiments to document interrelationship between proteasome-dependent potentiation and homeostatic control are carefully designed, with excellent, novel tools, selective pre- and post-synaptic expression of relevant perturbations and systematic analysis of all relevant synaptic parameters using the best possible methodology, with all the appropriate controls. The involvement of synaptic vesicles loosely coupled to Ca^{2+} -channels is convincingly demonstrated. The inhibitory effect of EGTA-AM during homeostatic scaling, but not under normal conditions, is remarkable. On the other hand, there are some issues concerning novelty, both regarding proteasome- and Dysbindin involvement in homeostatic scaling, and there is too little effort to connect to existing literature and reach consensus. Furthermore, the experiments to tie Dysbinding function into proteasome-dependent potentiation and homeostatic control are not as thorough as the experiments on the interrelationship between the two processes (Fig 1-6) and do not help to explain the data in the first 6 figures. Instead, the robust findings in the first 6 figures remain unexplained and the most crucial questions are not addressed. Together this makes the current version of the manuscript rather incoherent and unsatisfactory, with many open ends and too many topics addressed only partially. The title of the manuscript in fact applies only to the data in the last figure (Fig 8), which contains insufficient data to support this.

It would be better to split the current data set into 2 separate manuscripts, on (i) the interrelationship between homeostatic control and proteasome-dependent inhibition of EGTA-sensitive vesicles and (i)

the role of Dysbindin in regulating EGTA-sensitive vesicles; to dig deeper into each topic and connect better to existing literature. Fig 1-6 would make a great manuscript when 2-3 data sets were added to link the current data set to existing literature. The Dysbindin data set would require more work.

MAJOR ISSUES

1- Novelty & conceptual consensus

The involvement of the proteasome in homeostatic control of synaptic transmission is already well described in mammalian systems by the Fejtova lab and the Moulder lab. The current manuscript goes beyond what is already known in terms of selective manipulation and temporal control of the effects (both acute and chronic) and is also more convincing than published data. Unfortunately, there is little connection between the mechanisms described in mammalian and fly synapses, also little efforts to establish this, and generally there is a lack of overarching models to explain homeostatic control. For fly neuromuscular junction alone, the picture is already very complex: dozens of studies have implicated genes, pathways and biophysical principles in homeostatic control, but new studies, including the current one, fail to connect to the previous ones. The consequence is a conceptual zoo, where everyone gets lost except the scientists that work on this topic. In the end, this is not good for the field. This issue applies to both the proteasome-dependent potentiation and for its role homeostatic scaling:

1a- Proteasome-dependent potentiation

A previous study showed that in fly neuromuscular junction, proteasome inhibition enhances basic synaptic transmission. This is nicely confirmed in the current study. However, the previous study showed this is due to specific accumulation of presynaptic Dunc13. This would be consistent with an increase in RRP size (Fig 5). The current study tests several presynaptic genes (Fig 7), including Dunc13 binding protein RIM, but not Dunc13 itself. Why not? If Dunc13 is indeed upregulated upon proteasome inhibition (confirmation of this old finding is really necessary and important), it is striking that homeostatic plasticity is completely blocked at the same time. Does Dunc13 still accumulate after receptor blockade/deletion? What does Dunc13 overexpression do on basal transmission, on homeostatic plasticity and on proteasome-dependent potentiation? Since RIM is required for expression of proteasome-dependent potentiation, does RIM accumulate after Lac too? Does RIM-overexpression induce Dunc13 accumulation? Is RIM over-expression non-additive to proteasome inhibition? Do Dunc13 and RIM overexpression produce similar effects on RRP-size and calcium transients as proteasome inhibition? Are Dunc13 and RIM also required for DTS(pre)-dependent potentiation? Answers to such questions would help to explain the current data set (Fig 1-6) and to connect to existing literature.

1b- homeostatic scaling

Active zone matrix proteins Bassoon, Piccolo and Fife have been shown to regulate homeostatic control of synaptic transmission and vesicular release probability at the same time, the first two also in relation proteasome inhibition. Why is regulation of these proteins (or their fly orthologs) not tested (same questions as above)?

If proteasome-dependent potentiation is not expressed during homeostatic scaling (Fig 2), the obvious question is: does homeostatic scaling inhibit the presynaptic proteasome (as also proposed in the authors' final model, Fig 8h)? Reporters/assays are available to answer this obvious question directly. In addition to the proteins mentioned above, at least a dozen presynaptic proteins have previously been shown to be essential for homeostatic control (an auxiliary subunit of the calcium channel, an innate immune receptor, Endostatin, an ENaC channel, C-terminal SRC kinase, Rab3-GAP, Snapin, MCTP, a ER-calcium sensor etc. etc.). At least some efforts should be made (at least in the Discussion) to reconcile all these findings and connect them to the data in this manuscript.

2- Dysbinding function in proteasome-dependent regulation and homeostatic plasticity

The experiments to elucidate the presynaptic mechanisms proteasome-dependent regulation of

synaptic transmission (last 2 figures) are not as thorough as the first 6 figures. The role of Dysbindin is interesting and convincing, especially because Dysbindin loss does not have a major impact on basal transmission (unlike other proteins implicated homeostatic plasticity like RIM and calcium channels), but not really new (at least for homeostatic plasticity). The link with the first 6 figures is weak and new mechanistic insights are limited.

A thorough analysis of Dysbindin function in regulating (low probability) synaptic vesicle availability and/or fusion would require morphological analysis of synaptic vesicles at the ultrastructural level and attempts to assess how/where Dysbindin function connects to proteasome-dependent potentiation and homeostatic scaling: does Dysbindin accumulate at the terminal after proteasome inhibition? Does Dysbindin overexpression also increase RRP-size and calcium transients like proteasome inhibition does? Does Dysbindin overexpression accumulates Dunc13 and/or RIM and vice versa, etc.

Minor issues:

-It is not clear how mutant proteasome subunit DTS5 was inhibited. Larvae were probably placed from permissive to non-permissive temperature (when? how long?)

-Title: the data contain good evidence that the relevant proteasome activity is presynaptic. This should be expressed in the title

-units are lacking in Fig 6f (probably pC)

Reviewer #3 (Remarks to the Author):

This ms examines the role of proteasome function in presynaptic homeostatic plasticity at the NMJ of drosophila, a system that has been well studied in the recent years in regards of homeostatic plasticity mechanisms.

Here the authors quite convincingly show that induction of presynaptic homeostatic plasticity is blocked after acute pharmacological or prolonged genetic inhibition of presynaptic proteasome function, and inhibition of proteasome function rapidly potentiates synaptic transmission. Analysis of the potentiated state showed multiple effects, including higher Ca influx, an apparently increased vesicle pool size, slowing of EPSC kinetics and enhanced sensitivity to EGTA. This phenomenon is dependent on dysbindin. The data are generally of high quality and are well presented. I find the main conclusion that a dysbindin sensitive pool of loosely Ca coupled vesicle is the major component that is modulated in this form of homeostatic plasticity and is particularly sensitive to proteasome function well supported by the experimental data. This ms is a substantial contribution to the current knowledge of presynaptic homeostatic plasticity at fruit fly NMJ.

I have only minor comments, criticisms:

1. Fig 6 and discussion line 543 onwards. The change in RRP may well be not real, and the result of a primary alteration in Ca channel function. The calculation of the apparent RRP size is based on trains of Ca evoked responses, see for example Lou et al., J Neurosci. 2008. So this possibility should be more explicitly discussed.

2. Staining for SK2 is in not convincing. Please provide more examples for data shown in S1 d or some control data that the labeling is specific.

Reviewer #4 (Remarks to the Author):

Synapses require flexibility (i.e. plasticity) to adjust to environmental challenges. Yet there must also exist regulatory mechanisms that constrain activity within appropriate physiological ranges. An abundance of evidence suggests that homeostatic regulation is critical in this regard. The *Drosophila melanogaster* third instar larval neuromuscular junction (NMJ) has developed into an important

experimental system in this regard, and homeostatic potentiation has been systematically investigated using predominantly genetics. To study short-term regulation, Philantotoxin acutely blocking postsynaptic glutamate receptors is generally used, while genetic elimination of the *glurIIA* receptor subunit is generally used to study chronic compensation.

In this manuscript, the relationship between homeostatic plasticity and presynaptic proteasome function is investigated at the larval *Drosophila* NMJ. The authors interfere with proteasome function by using i) an application of 100 μ M lactacystin for different time durations, or ii) a motoneuron expression of temperature sensitive versions of proteasome subunits (DTS). Both "proteasome perturbation" constellations they find to increase baseline transmission, and "block" the quantal content upregulation normally occurring after application of Philantotoxin (PhTx). Increased SV release they found associated with increased Ca^{2+} influx (via OGB-1 fluorescence imaging) and increased EGTA sensitivity, indicative of the additionally recruited SVs being rather loosely coupled. They analyse the proteasome perturbations experiments in the background of "chronic potentiation" as well, using genetic elimination of a specific glutamate receptor subunit (*GluRIIA*), finding similar "occlusion relations" as after acute proteasome perturbation. Using TEVC measurements, they report a "slowing of EPSC decay kinetics", which given that miniature events showed no such slowing, is likely of presynaptic character. Using EGTA buffer experiments, they provide evidence that both DTS-pre as well as PhTx treatment render a higher proportion of SV release EGTA sensitive than in controls. They then show that both, Ca^{2+} influx in confocal imaging is increased, and that RRP size (based on cumulative evoked amplitude plots) is elevated as well. Next, they analyse potentiation post proteasome perturbation in a spectrum of mutants previously shown to interfere with PhTx mediated potentiation of quantal content. *Dysbindin* as well as *RIM* mutants occlude proteasome perturbation mediated potentiation. Finally, they in more detail address the role of *Dysbindin*, showing that *Dysbindin* overexpression increases baseline release, and that after PhTx EGTA effects are occluded in *dysbinding* mutants, taking as evidence for *Dysbindin* promoting the recruitment of SVs at PhTx challenged NMJ terminals.

This manuscript provides a collection of interesting data concerning the mechanisms of SV recruitment at synapses under homeostatic challenge. Thus, it in my eyes is a promising candidate for publication in *Nature Communications*. However, this reviewer is not fully convinced that the findings presented, proteasome dependent up-regulation and EGTA-sensitive SVs being recruited under homeostatic challenge, necessarily needed to be combined within one manuscript. That said, this reviewer principally supports publication, but would like to await their response to the following points.

POINTS:

1. PhTx application subsequent to proteasome blockade failed to provoke further increases of quantal content: "blockade of homeostatic potentiation" after proteasome blockade. They concluded: "These data demonstrate that proteasome function is required for the rapid induction of presynaptic homeostatic plasticity". However: doesn't the system just have a saturation ("ceiling") point concerning quantal content? Could they provide examples for homeostatic potentiation to still operate under conditions of increased base-line transmission? They themselves mention a potential "ceiling effect". Would it make sense to test the interaction between both treatments over a range of Ca^{2+} concentrations, e.g.? I do not fully support their statements here ("Conversely, an absence of an effect on release would indicate that both perturbations enhance release through similar mechanisms").
2. How exactly did they perform their DTS proteasome experiments? Which temperature? Did they need to upshift temperature for effects? Somehow I could not find this information in the legends or methods section.
3. How do they interpret that overshoot of evoked release after PhTx application (Fig. 1B)
4. Mini amplitudes appeared clearly increased after muscle expression of DTS but not after presynaptic

(motoneuron) expression. Are glutamate receptor levels increased here?

5. Morphological analysis: the size of individual presynaptic active zones might change...did they check? How about glutamate receptor field size?

6. "This decrease was less pronounced than the EGTA-induced reduction in EPSC amplitude of PhTX324 treated control synapses, indicating that control synapses are more EGTA sensitive during homeostatic plasticity than DTSpres mutant synapses". They don't use statistics to underline this statement.

7. Could they plot quantal content data for figure 5?

8. They should take reference to earlier work at the NMJ using EGTA-AM esters. They anyway strongly prefer to refer to their work and homeostatic work of the Davis lab. A more balanced referencing and discussion of also other work concerning the elucidation of presynaptic release machinery at the NMJ would be appreciated. This also applies to recent work elucidating mechanisms of loose versus tight coupling in this system.

9. Could they perform RRP estimates under EGTA-AM?

10. Given that the senior author was involved in work implicating Rim-binding protein in the homeostatic scaling, I think they should investigate the role of rim-bp for plasticity after proteasome perturbation.

11. usage of the term synapse: a whole NMJ terminal is "a synapse" obviously in their terminology. They should clarify this.

12. Dlg is not a postsynaptic DENSITY marker here but labels the whole membrane of the postsynaptic reticulum

Resonse to the reviewers' comments

NCOMMS-17 10144

Reviewers' comments:

Reviewer #1 (Remarks to the Author):

This paper examines presynaptic mechanisms underlying homeostatic plasticity at the *Drosophila* NMJ. This preparation has proven to be a powerful system in which to study homeostatic signaling at synapses, and has been used by several groups to identify genetic and molecular determinants of synaptic compensation. Here, the authors primarily focus on the presynaptic role of the proteasome, and argue that presynaptic protein degradation opposes the release of "loosely coupled," low-release probability vesicles that are recruited during homeostatic regulation of quantal content. They go on to conclude that the gene *Dysbindin*, which has previously been implicated in presynaptic homeostasis at the fly NMJ, plays a key role in regulating access to these "loosely-coupled" vesicles. As such, the authors do a nice job of integrating their novel findings within a mechanistic framework that has been established by the powerful genetics of this system.

Overall, the experiments have been thoughtfully executed with appropriate controls. The findings reveal novel roles of proteasome-dependent degradation and *Dysbindin* in homeostasis and are likely to be of broad interest to those in the field. However, there are several issues that need to be addressed prior to publication:

MAJOR CONCERNS

1) A moderate concern related to the approach is that proteasome disruption, particularly chronic proteasome disruption, is likely to have numerous secondary effects that profoundly alter cell physiology. This can complicate the rather simple interpretation that degradation of some pool of presynaptic proteins is directly tied to vesicle release. The authors are quite up front about this issue, and do a reasonable job of addressing it – for example, using the genetic strategy to complement the pharmacological inhibitors. However, it would be nice to see a little more validation of the main effects using other proteasome inhibitors (e.g., epoximycin, MG132), and these experiments seem fairly straightforward. In addition, the authors should probably weigh the secondary impact of proteasome inhibition a little more strongly in their interpretation of a role for protein degradation specifically, in presynaptic homeostasis. For example, it is well known that strong proteasome inhibition can deplete the free ubiquitin pool in cells, which can impact other cellular events controlled by mono-ubiquitination. Some discussion of this issue is warranted.

- We would like to thank reviewer #1 for the valuable suggestions and comments. We conducted two additional experiments to address this first concern: First, we used the specific proteasome blocker MG-132 and found that MG-132 application enhanced release and completely blocked homeostatic potentiation of release (now added to Figure 1), very similar to lactacystin treatment or presynaptic DTS overexpression. Second, we probed homeostatic plasticity in cut-up (ctp) mutants, which were recently shown to reduce the number and mobility of proteasomes at the *Drosophila* NMJ due to a defect in proteasome trafficking (Kreko-Pierce and Eaton, 2017; ctp encodes a dynein light chain subunit required for proteasome trafficking). Homeostatic potentiation of release was blocked in ctp mutants (Figure S2). Together, four independent experimental perturbations targeting the proteasome (i.e., lactacystin, MG-132, DTS overexpression and ctp) induce a very similar phenotype in synaptic physiology, thereby further strengthening our data and conclusions. However, we also discuss possible secondary effects of proteasome perturbation on cellular physiology including mono-ubiquitination and ubiquitin levels (page 24, line 576).

2) An important tool used throughout the paper is the genetic inhibition of the proteasome through expression of mutant proteasome subunits. This tool is the basis for the presynaptic role of the proteasome the authors identify, which is a central finding of the paper. Surprisingly, however, the authors provide little in the way of rationale for this approach. On page 7, they describe the basic strategy – overexpression of one or two dominant temperature-sensitive mutant proteasome subunits – but do not articulate why this strategy impairs proteasome function. They refer the reader to the methods, but very little information is provided there either. It is not clear why expression of these mutant subunits impairs proteasome function, and no mention is made of shifting temperatures as is standard with temperature-sensitive mutants. Given the central importance of this tool to the conclusions, the authors must provide more information regarding this approach, and also provide data demonstrating how effectively this strategy achieves the desired effect – i.e., how strongly proteasome function is disrupted in flies expressing these mutant subunits. Much more needs to be done here.

- We now describe how DTS overexpression is thought to inhibit proteasome function in the results section (page 7, line 147). In brief, based on the results described by Smyth and Belote (1999) and Belote and Fortier (2002), DTS overexpression impairs proteasome function because of single-amino-acid substitutions in the $\beta 6$ subunit and $\beta 2$ subunit of the 20S proteasome, which results in misfolding of the β -subunits.

- We also carried out additional experiments (Figures 7d, e, S2c, d) showing that the effects of DTS expression on synaptic physiology depend on DTS copy number. Specifically, we observed that presynaptic overexpression of one DTS copy is not sufficient to potentiate release

under baseline conditions (Figure S2c, d), or to block homeostatic plasticity (Figure 7d, e). Similar results were obtained upon DTS expression at lower temperatures (data not shown). We conclude that DTS expression affects synaptic physiology in a dose-dependent manner.

- These observations are in line with previous work demonstrating that DTS expression disrupts proteasome function in the *Drosophila* eye in a dosage-dependent manner (Smyth and Belote, 1999). This earlier study also revealed that the degree of proteasome perturbation is positively correlated with the temperature at which DTS was expressed throughout development. Together, overexpression of DTS is thought to lead to "hypomorphic", partial loss of proteasome function that depends on DTS copy number and temperature.

- Based on this previous work, we overexpressed one or two DTS copies at a permissive temperature of 25°C throughout development. This is now described on page 7 (line 147) and on page 29 (line 685). We also confirmed that the expression of two DTS copies at higher temperatures (29°C) throughout development results in a stronger phenotype (lethality at the pupal stage; data not shown).

- It is also worth noting that DTS had already been validated at the *Drosophila* NMJ in a previous study (Speese et al., 2003). This study revealed that presynaptic DTS expression results in elevated synaptic unc-13 levels.

3) The authors define a pool of synaptic vesicles as "loosely-coupled," due to the fact that the slow Ca²⁺ chelator EGTA apparently reduces their release. The logic is that slow chelators do not abolish release because the site of Ca²⁺ entry is so close to the primed vesicles, which are "tightly coupled" to Ca²⁺ influx. By extension, then, the interpretation is that vesicles sensitive to EGTA must be more loosely coupled to the site of Ca²⁺ entry. However, there are numerous Ca²⁺-dependent events (e.g., priming), aside from direct vesicle fusion, that regulate transmission from presynaptic terminals, and EGTA could be impacting these. And slow chelators like EGTA have been shown time-and-time again to disrupt presynaptic short-term plasticity such as facilitation, PTP, and augmentation that arise due to Ca²⁺-dependent changes in release probability. Moreover, the authors demonstrate that basal Ca²⁺ levels are increased with proteasome inhibition, suggesting that the EGTA-dependent effects they observe may simply be due to reducing free Ca²⁺ independent of fusion. Given these issues, it is important for the authors to provide other evidence (aside from EGTA sensitivity) for this presumably distinct vesicle pool.

- We addressed this question with two new experiments that do not involve EGTA application: First, we investigated recovery kinetics after vesicle pool depletion. Work from various synapses has demonstrated a bi-exponential recovery time course following pool depletion, and it is thought that the fast recovery component reflects the replenishment of

vesicles with a low release probability (reviewed by Hallermann and Silver, 2013). Presynaptic DTS expression results in enhanced recovery during the fast recovery phase (Figure 6g, h), providing indirect evidence for enhanced recruitment of vesicles with a low release probability. To our knowledge, there is no experimental evidence that changes in presynaptic calcium influx (e.g. by changing the extracellular calcium concentration) result in a similar effect.

- Second, based on the finding that presynaptic DTS overexpression induces a significant slowing of EPSC decay kinetics (Figure 4), we tested if a similar phenotype is observed upon increasing the extracellular calcium concentration. We therefore recorded EPSCs at elevated extracellular calcium concentration (2mM) and quantified EPSC decay kinetics. There are no significant changes in EPSC decay kinetics between control synapses recorded at 1mM and 2mM extracellular calcium (Figure S5c), indicating that an increase in presynaptic calcium influx alone is unlikely to underlie the slowing of EPSC decay kinetics after presynaptic DTS expression (Figure 4).

- Third, we did not observe major changes in the time course or degree of synaptic short-term depression during stimulus trains after proteasome perturbation (Figure S5b) or dysbindin overexpression (Figure 8), suggesting that these manipulations predominately potentiate RRP size rather than inducing major changes in release probability.

- Finally, there is evidence that dysbindin regulates release independent of presynaptic calcium influx (Dickman and Davis, 2009). We now provide evidence that presynaptic dysbindin overexpression increases vesicle pool size (Figure 8f, g), and that this pool is more EGTA sensitive (Figure S5e). Together, these experiments, which are mostly independent of EGTA application, support the hypothesis that the observed effects of proteasome perturbation on release are not only caused by increased presynaptic calcium influx, thereby pointing towards the recruitment of a vesicle pool with a lower release probability.

- To further clarify this issue, we toned down our interpretation of the EGTA data throughout the results section and mostly replaced 'loosely-coupled' by 'EGTA sensitive', thereby emphasizing the experimental result rather than its interpretation (e.g p.15, l.328). Moreover, in the discussion we state that "The increase in RRP size or EGTA-sensitivity seen upon proteasome inhibition may therefore be in part a secondary consequence of enhanced presynaptic Ca^{2+} influx." (p. 27, l. 652).

4) The experiments provide little in the way of mechanistic insight into how Dysbindin recruits the novel population of low Pr vesicles. Dysbindin has already been established as an essential homeostatic plasticity gene, so the fact that it regulates release and EGTA sensitivity is not altogether surprising. Are the levels of Dysbindin altered during homeostatic plasticity, and is this mostly via a change in half life as might be expected for a target regulated by the proteasome? By what mechanism does Dysbindin control the availability of the low Pr vesicle pool? The paper

would be stronger if the authors could provide more data what speak directly to these questions.

- Unfortunately, there is currently no available antibody to probe dysbindin levels or dysbindin half-life in *Drosophila* (the same applies for RIM, the only other gene we could genetically link to proteasome function). We tested an antibody kindly provided by Dion Dickman's lab, but both labs failed to detect endogenous dysbindin at the NMJ using this antibody. We attempted quantifying the fluorescence intensity of venus-tagged dysbindin that was overexpressed presynaptically. However, the fluorescence intensity, which may not reflect endogenous dysbindin levels, was too heterogeneous within a given NMJ and between NMJs to resolve potential differences.

- However, our genetic data suggest that dysbindin levels are important, because dysbindin overexpression increases vesicle pool size (Figure 8).

- Several studies have addressed the role of dysbindin in homeostatic regulation of release, but the underlying mechanisms have remained largely elusive. As mentioned in the introduction, there is evidence that dysbindin regulates release in concert with *snarin* and SNAP-25 during synaptic homeostasis (Dickman et al., 2012) downstream of presynaptic calcium influx (Dickman and Davis, 2009). Recent genetic data suggest that *dysbindin*-dependent homeostatic regulation of release involves the BLOC-1 complex member *Blos1* (Mullin et al., 2015) and the Arp2/3 Actin Polymerization Complex (Gokhale et al., 2016). Future work will address if and how some of these genes are involved in recruiting synaptic vesicles during homeostatic plasticity or after proteasome function.

SPECIFIC COMMENTS

1) The authors discuss (on page 6) how their results on the effects of proteasome inhibition extend previous work from the *Drosophila* NMJ. There is a fairly substantial literature on this from mammalian central synapses as well that should be included here.

- We included this information along with five references to the introduction (page 3, line 65).

2) The authors do a nice job of showing raw values for EPSP and mEPSP amplitude, in addition to the computed changes in quantal content for all of their manipulations. However, they never mention how mEPSP frequency is altered in these experiments? I realize that many papers fail to mention mEPSP frequency, but this is another source of information that is clearly relevant to presynaptic mechanisms. Especially given recent evidence that evoked and spontaneous release may be uniquely regulated, it is worth providing this data.

- We now show mEPSP frequency data in Figure S2a and b. There are no apparent changes in mEPSP frequency upon genetic proteasome perturbation. It is worth noting that homeostatic plasticity also does not affect mEPSP frequency at this synapse (Figure S2a and b). Moreover, the

increase in release probability during evoked release does not necessarily translate into changes in mEPSP frequency at this synapse.

Reviewer #2 (Remarks to the Author):

This study addresses contributions of the presynaptic proteasome to homeostatic control of synaptic transmission at the *Drosophila* neuromuscular junction. The interrelationship between proteasome-dependent potentiation and homeostatic control is documented in detail (6 figures), followed by experiments to link two presynaptic proteins, RIM and especially Dysbindin, to these two processes.

The experiments to document interrelationship between proteasome-dependent potentiation and homeostatic control are carefully designed, with excellent, novel tools, selective pre- and post-synaptic expression of relevant perturbations and systematic analysis of all relevant synaptic parameters using the best possible methodology, with all the appropriate controls. The involvement of synaptic vesicles loosely coupled to Ca²⁺-channels is convincingly demonstrated. The inhibitory effect of EGTA-AM during homeostatic scaling, but not under normal conditions, is remarkable. On the other hand, there are some issues concerning novelty, both regarding proteasome- and Dysbindin involvement in homeostatic scaling, and there is too little effort to connect to existing literature and reach consensus. Furthermore, the experiments to tie Dysbinding function into proteasome-dependent potentiation and homeostatic control are not as thorough as the experiments on the interrelationship between the

two processes (Fig 1-6) and do not help to explain the data in the first 6 figures. Instead, the robust findings in the first 6 figures remain unexplained and the most crucial questions are not addressed. Together this makes the current version of the manuscript rather incoherent and unsatisfactory, with many open ends and too many topics addressed only partially. The title of the manuscript in fact applies only to the data in the last figure (Fig 8), which contains insufficient data to support this.

It would be better to split the current data set into 2 separate manuscripts, on (i) the interrelationship between homeostatic control and proteasome-dependent inhibition of EGTA-sensitive vesicles and (ii) the role of Dysbindin in regulating EGTA-sensitive vesicles; to dig deeper into each topic and connect better to existing literature. Fig 1-6 would make a great manuscript when 2-3 data sets were added to link the current data set to existing literature. The Dysbindin data set would require more work.

- After adding new data and discussing this issue with the editor, we decided against splitting the manuscript.

MAJOR ISSUES

1- Novelty & conceptual consensus

The involvement of the proteasome in homeostatic control of synaptic transmission is already well described in mammalian systems by the Fejtova lab and the Moulder lab. The current manuscript goes beyond what is already known in terms of selective manipulation and temporal control of the effects (both acute and chronic) and is also more convincing than published data. Unfortunately, there is little connection between the mechanisms described in mammalian and fly synapses, also little efforts to establish this, and generally there is a lack of overarching models to explain homeostatic control. For fly neuromuscular junction alone, the picture is already very complex: dozens of studies have implicated genes, pathways and biophysical principles in homeostatic control, but new studies, including the current one, fail to connect to the previous ones. The consequence is a conceptual zoo, where everyone gets lost except the scientists that work on this topic. In the end, this is not good for the field. This issue applies to both the proteasome-dependent potentiation and for its role homeostatic scaling:

1a- Proteasome-dependent potentiation

A previous study showed that in fly neuromuscular junction, proteasome inhibition enhances basic synaptic transmission. This is nicely confirmed in the current study. However, the previous study showed this is due to specific accumulation of presynaptic Dunc13. This would be consistent with an increase in RRP size (Fig 5). The current study tests several presynaptic genes (Fig 7), including Dunc13 binding protein RIM, but not Dunc13 itself. Why not? If Dunc13 is indeed upregulated upon proteasome inhibition (confirmation of this old finding is really necessary and important), it is striking that homeostatic plasticity is completely blocked at the same time. Does Dunc13 still accumulate after receptor blockade/deletion? What does Dunc13 overexpression do on basal transmission, on homeostatic plasticity and on proteasome-dependent potentiation? Since RIM is required for expression of proteasome-dependent potentiation, does RIM accumulate after Lac too? Does RIM-overexpression induce Dunc13 accumulation? Is RIM over-expression non-additive to proteasome inhibition? Do Dunc13 and RIM overexpression produce similar effects on RRP-size and calcium transients as proteasome inhibition? Are Dunc13 and RIM also required for DTS(pre)-dependent potentiation? Answers to such questions would help to explain the current data set (Fig 1-6) and to connect to existing literature.

- We would like to thank the reviewer for the valuable comments. Based on this first suggestion, we tested recently published *unc13A* and *unc13B* *Drosophila* mutants (Böhme et al., 2016) in the context of homeostatic plasticity. Interestingly, we found no defect in homeostatic plasticity in both mutants (these data are part of another manuscript). Based on the focus of the current manuscript on links between homeostatic plasticity and proteasome function, these findings kept us from continuing to investigate *unc13* in the context of the present study.

- We decided to further link dysbindin to homeostatic plasticity and proteasome function instead of repeating the suggested experiments for RIM. It is worth noting that it would have been equally challenging to link RIM to proteasome function, because a specific antibody that would permit investigating RIM levels at the *Drosophila* NMJ is currently not available (see point 4 of reviewer 1).

1b- homeostatic scaling

Active zone matrix proteins Bassoon, Piccolo and Fife have been shown to regulate homeostatic control of synaptic transmission and vesicular release probability at the same time, the first two also in relation proteasome inhibition. Why is regulation of these proteins (or their fly orthologs) not tested (same questions as above)?

If proteasome-dependent potentiation is not expressed during homeostatic scaling (Fig 2), the obvious question is: does homeostatic scaling inhibit the presynaptic proteasome (as also proposed in the authors' final model, Fig 8h)? Reporters/assays are available to answer this obvious question directly.

In addition to the proteins mentioned above, at least a dozen presynaptic proteins have previously been shown to be essential for homeostatic control (an auxiliary subunit of the calcium channel, an innate immune receptor, Endostatin, an ENaC channel, C-terminal SRC kinase, Rab3-GAP, Snapin, MCTP, a ER-calcium sensor etc. etc.). At least some efforts should be made (at least in the Discussion) to reconcile all these findings and connect them to the data in this manuscript.

- We now studied the role of *Drosophila* Fife (Piccolo-RIM-related) in the context of proteasome function (Figure 7). Pharmacological proteasome inhibition for 15 minutes produced a significant increase in EPSP amplitude in *fife* mutants (Figure 7b), suggesting that proteasome-dependent regulation of Fife does not result in significant changes in synaptic transmission under our experimental conditions.

- We tested two reporters (CL-1, Pandey et al., 2007; and DHFRts, Speese et al., 2003) to address this issue. However, both CL-1 and DHFRts fluorescence levels were too low under different experimental conditions to allow for a quantitative analysis of proteasome function during homeostatic plasticity (data not shown).

- We performed new experiments to strengthen the link between proteasome function and homeostatic plasticity. We uncovered a block in homeostatic plasticity in *cut-up* (*ctp*) mutants (Figure S2e,f), which were recently shown to have reduced synaptic proteasome number and mobility (Kreko-Pierce and Eaton, 2017; *ctp* encodes the dynein light chain subunit LC8). This finding opens up the possibility that proteasome mobility and/or recruitment may be modulated during homeostatic plasticity. We now discuss this possibility (p.25 l.603) and actively investigate a possible role of proteasome mobility during homeostatic plasticity, which we consider to be beyond the scope of the present study.

- We investigated rab3-GAP, ppk16 (encoding an ENaC channel) and seven further presynaptic homeostasis mutants in the context of proteasome function. All mutants but dysbindin and rim mutants displayed an increase in EPSP amplitude after pharmacological proteasome inhibition. We further highlight these observations in the discussion (p.25, l.592).

2- Dysbinding function in proteasome-dependent regulation and homeostatic plasticity

The experiments to elucidate the presynaptic mechanisms proteasome-dependent regulation of synaptic transmission (last 2 figures) are not as thorough as the first 6 figures. The role of Dysbindin is interesting and convincing, especially because Dysbindin loss does not have a major impact on basal transmission (unlike other proteins implicated homeostatic plasticity like RIM and calcium channels), but not really new (at least for homeostatic plasticity). The link with the first 6 figures is weak and new mechanistic insights are limited.

A thorough analysis of Dysbindin function in regulating (low probability) synaptic vesicle availability and/or fusion would require morphological analysis of synaptic vesicles at the ultrastructural level and attempts to assess how/where Dysbindin function connects to proteasome-dependent potentiation and homeostatic scaling: does Dysbindin accumulate at the terminal after proteasome inhibition? Does Dysbindin overexpression also increase RRP-size and calcium transients like proteasome inhibition does? Does Dysbindin overexpression accumulates Dunc13 and/or RIM and vice versa, etc.

We conducted three new experiments to further connect dysbindin to proteasome function and homeostatic plasticity.

1. We quantified vesicle pool size after presynaptic dysbindin overexpression and revealed that this manipulation increases readily-releasable vesicle pool (RRP) size (Figure 8f, g).
2. We probed EGTA-sensitivity of the vesicle pool that is recruited upon dysbindin overexpression. Consistent with our hypothesis, we observed a significantly increased EGTA-sensitivity of the RRP after dysbindin overexpression (Figure S5e). Together, these data indicate that dysbindin levels control the size of an EGTA-sensitive vesicle pool.
3. We probed homeostatic plasticity in 'transheterozygous' mutants lacking one copy of dysbindin and expressing one DTS copy. Whereas heterozygous dysbindin mutants or expression of one DTS copy alone do not block homeostatic plasticity, homeostatic potentiation of release was impaired in transheterozygous mutants (Figure 7d, e). These data provide evidence for a genetic interaction between dysbindin and DTS during homeostatic plasticity.

Unfortunately, it is currently impossible to assay dysbindin levels or localization, because a specific antibody is lacking (We tested an antibody kindly provided by Dion Dickman's lab, but both labs failed to detect

dysbindin at the NMJ using this antibody, see point 4 by reviewer 1). We quantified the fluorescence intensity of venus-tagged dysbindin that was overexpressed presynaptically. However, the fluorescence intensity, which may not reflect endogenous dysbindin levels, was too heterogeneous within a given NMJ and between NMJs to resolve potential differences (See also point 4, reviewer 1).

Several studies have provided evidence that dysbindin functions downstream of presynaptic calcium influx, likely through snapin and SNAP-25 (Dickman and Davis, 2009; Dickman et al., 2012; Mullin et al., 2015; Gokhale et al., 2016). We therefore decided against investigating presynaptic calcium influx after dysbindin overexpression and discuss this issue on page 28 (line 664).

Minor issues:

-It is not clear how mutant proteasome subunit DTS5 was inhibited. Larvae were probably placed from permissive to non-permissive temperature (when? how long?)

- We now added this information to the results (page 7, line 147) and the methods section (page 29, line 685; for details see answer 2 to reviewer #1).

-Title: the data contain good evidence that the relevant proteasome activity is presynaptic. This should be expressed in the title .

- Thanks for this suggestion. We changed the title accordingly.

-units are lacking in Fig 6f (probably pC)

- RRP size was calculated by cum. EPSC charge (pC)/ miniature EPSC charge (pC) and is therefore unitless. We added this information to the figure legend of figure 6f.

Reviewer #3 (Remarks to the Author):

This ms examines the role of proteasome function in presynaptic homeostatic plasticity at the NMJ of drosophila, a system that has been well studied in the recent years in regards of homeostatic plasticity mechanisms.

Here the authors quite convincingly show that induction of presynaptic homeostatic plasticity is blocked after acute pharmacological or prolonged genetic inhibition of presynaptic proteasome function, and inhibition of proteasome function rapidly potentiates synaptic transmission. Analysis of the potentiated state showed multiple effects, including higher Ca influx, an apparently increased vesicle pool size, slowing of EPSC kinetics and enhanced sensitivity to EGTA. This phenomenon is dependent on dysbindin. The data are generally of high

quality and are well presented. I find the main conclusion that a dysbindin sensitive pool of loosely Ca coupled vesicle is the major component that is modulated in this form of homeostatic plasticity and is particularly sensitive to proteasome function well supported by the experimental data. This ms is a substantial contribution to the current knowledge of presynaptic homeostatic plasticity at fruit fly NMJ.

I have only minor comments, criticisms:

1. Fig 6 and discussion line 543 onwards. The change in RRP may well be not real, and the result of a primary alteration in Ca channel function. The calculation of the apparent RRP size is based on trains of Ca evoked responses, see for example Lou et al., J Neurosci. 2008. So this possibility should be more explicitly discussed.

We would like to thank the reviewer for the constructive input.

Albeit we cannot (and don't want to) rule out that the apparent increase in RRP size is a secondary consequence of increased calcium influx, there are several lines of evidence suggesting that the increase in RRP size is not primarily caused by enhanced presynaptic calcium influx:

1. We provide additional data demonstrating that presynaptic DTS expression results in an increased fast recovery component after RRP depletion (Figure 6g, h), suggesting an increased fraction of low release probability vesicles (which are thought to undergo rapid recovery; for a review see Hallermann and Silver, 2013). A phenotype like this is not seen upon increasing the extracellular calcium concentration (e.g. see Müller et al., 2015).
2. Increasing extracellular calcium concentration does not induce a similar increase in EGTA-sensitivity of release (Figure 3B in Müller et al., 2015), or changes in EPSC decay kinetics (Figure S5c) as seen after proteasome impairment (Figures 4, 5).
3. We did not detect apparent changes in short-term plasticity, an indirect proxy for potential changes in release probability, upon proteasome inhibition (Figure S5b).
4. Overexpression of dysbindin, which is thought to act downstream of presynaptic calcium influx (Dickman and Davis, 2009; Dickman et al., 2012; Mullin et al., 2015; Gokhale et al., 2016), induces an increase in RRP size (Figure 8f, g) along with an increased EGTA-sensitivity of the RRP (Figure S5e).

Together, these data argue that the observed increase in apparent RRP size is not primarily caused by an increase in presynaptic calcium influx/release probability. We now dedicated a section of the discussion to this issue stating that the RRP size increase observed upon proteasome impairment may be partially due to an increase in presynaptic calcium influx (p. 27, l. 652; see also comment 3, reviewer #1).

2. Staining for SK2 is in not convincing. Please provide more examples for data shown in S1 d or some control data that the labeling is specific.

- We added more examples to figure S1. Moreover, we normalized the FK2 fluorescence intensity data to the intensity of the HRP mask that was used to segment the presynaptic compartment. We found a significant increase in the FK2/HRP fluorescence intensity ratio upon lactacystin treatment (Figure S1e), similar to the increase in FK2 fluorescence intensity alone (Figure S1e, f). To further illustrate the increase in FK2 fluorescence intensity after lactacystin application, we added a cumulative frequency plot of the FK2 data in the absence and presence of lactacystin (Figure S1g).

Reviewer #4 (Remarks to the Author):

Synapses require flexibility (i.e. plasticity) to adjust to environmental challenges. Yet there must also exist regulatory mechanisms that constrain activity within appropriate physiological ranges. An abundance of evidence suggests that homeostatic regulation is critical in this regard. The *Drosophila melanogaster* third instar larval neuromuscular junction (NMJ) has developed into an important experimental system in this regard, and homeostatic potentiation has been systematically investigated using predominantly genetics. To study short-term regulation, Philantotoxin acutely blocking postsynaptic glutamate receptors is generally used, while genetic elimination of the *glurIIA* receptor subunit is generally used to study chronic compensation.

In this manuscript, the relationship between homeostatic plasticity and presynaptic proteasome function is investigated at the larval *Drosophila* NMJ. The authors interfere with proteasome function by using i) an application of 100 μ M lactacystin for different time durations, or ii) a motoneuron expression of temperature sensitive versions of proteasome subunits (DTS). Both "proteasome perturbation" constellations they find to increase baseline transmission, and "block" the quantal content upregulation normally occurring after application of Philantotoxin (PhTx). Increased SV release they found associated with increased Ca^{2+} influx (via OGB-1 fluorescence imaging) and increased EGTA sensitivity, indicative of the additionally recruited SVs being rather loosely coupled. They analyse the proteasome perturbations experiments in the background of "chronic potentiation" as well, using genetic elimination of a specific glutamate receptor subunit (*GluRIIA*), finding similar "occlusion relations" as after acute proteasome perturbation. Using TEVC measurements, they report a "slowing of EPSC decay kinetics", which given that miniature events showed no such slowing, is likely of presynaptic character. Using EGTA buffer experiments, they provide evidence that both DTS-pre as well as PhTox treatment render a higher proportion of SV release EGTA sensitive than in controls. They then show that both, Ca^{2+} influx in confocal imaging is increased, and that RRP size (based on cumulative evoked amplitude plots) is elevated as well. Next,

they analyse potentiation post proteasome perturbation in a spectrum of mutants previously shown to interfere with PhTox mediated potentiation of quantal content. Dysbindin as well as RIM mutants occlude proteasome perturbation mediated potentiation. Finally, they in more detail address the role of Dysbindin, showing that Dysbindin overexpression increases baseline release, and that after PhTox EGTA effects are occluded in dysbinding mutants, taking as evidence for Dysbindin promoting the recruitment of SVs at PhTox challenged NMJ terminals.

This manuscript provides a collection of interesting data concerning the mechanisms of SV recruitment at synapses under homeostatic challenge. Thus, it in my eyes is a promising candidate for publication in Nature Communications. However, this reviewer is not fully convinced that the findings presented, proteasome dependent up-regulation and EGTA-sensitive SVs being recruited under homeostatic challenge, necessarily needed to be combined within one manuscript. That said, this reviewer principally supports publication, but would like to await their response to the following points.

POINTS:

1. PhTox application subsequent to proteasome blockade failed to provoke further increases of quantal content: "blockade of homeostatic potentiation" after proteasome blockade. They concluded: "These data demonstrate that proteasome function is required for the rapid induction of presynaptic homeostatic plasticity". However: doesn't the system just have a saturation ("ceiling") point concerning quantal content? Could they provide examples for homeostatic potentiation to still operate under conditions of increased base-line transmission? They themselves mention a potential "ceiling effect". Would it make sense to test the interaction between both treatments over a range of Ca²⁺ concentrations, e.g.? I do not fully support their statements here ("Conversely, an absence of an effect on release would indicate that both perturbations enhance release through similar mechanisms").

We would like to thank the reviewer for the valuable comments and suggestions.

- As suggested by the reviewer, we investigated the effect of proteasome perturbation on homeostatic plasticity under conditions of increased baseline synaptic transmission. Genetic proteasome perturbation completely blocks homeostatic plasticity at an extracellular calcium concentration of 1mM (Figures 4, 5), very similar to 0.3mM (Figure 1).
- Furthermore, other independent data sets, such as the comparison between PhTX-treated control synapses (quantal content = 48; Figure 1b) and lactacystin-treated synapses without glutamate receptor perturbation (quantal content = 33), argue against a ceiling effect. Thus, although we cannot fully exclude a saturation/ceiling effect, these data indicate that

saturation is unlikely the major cause underlying the defect in homeostatic plasticity upon proteasome inhibition. We now clearly state this in the results "Although we cannot exclude a ceiling effect, we consider this unlikely because release was not saturated after lactacystin application under our recording conditions (quantal content ~35 vs. ~50 after PhTX treatment; Figure 1b), and because homeostatic plasticity was also completely blocked after proteasome perturbation at elevated extracellular Ca^{2+} concentrations (1mM Figure 4)" (page 7, line 127).

- Concerning the statement "Conversely, an absence of an effect on release would indicate that both perturbations enhance release through similar mechanisms": We agree and changed this statement to "By contrast, an absence of an effect on release would indicate that both perturbations are non-additive." (page 10, line 205)

2. How exactly did they perform their DTS proteasome experiments? Which temperature? Did they need to upshift temperature for effects? Somehow I could not find this information in the legends or methods section.

- We now described the DTS experiments in more detail in the results and the methods sections (page 7, line 147; and page 29, line 685, respectively; see also point 2, reviewer #1 for a detailed description).

3. How do they interpret that overshoot of evoked release after PhTox application (Fig. 1B)

- The overshoot in EPSP amplitude upon PhTX application in wild type of the original data set was not statistically significant. However, we increased sample size and the EPSP in the absence and presence of PhTX are similar in the new data set (Figure 1b).

4. Mini amplitudes appeared clearly increased after muscle expression of DTS but not after presynaptic (motoneuron) expression. Are glutamate receptor levels increased here?

- A previous study investigated this phenotype in greater detail and observed an increase in GluRIIB-containing receptor levels (Haas et al., 2007). Additionally, we now imaged GluR levels upon presynaptic DTS overexpression and did not detect apparent differences in GluRIIA levels (Figure S4). We focused on GluRIIA levels because presynaptic DTS expression accelerated mEPSC decay kinetics indicative of potential changes in GluRIIA levels.

5. Morphological analysis: the size of individual presynaptic active zones might change...did they check? How about glutamate receptor field size?

- We now analyzed Brp puncta size and GluR field size and did not detect apparent differences between controls and presynaptic DTS mutants (Figure S4).

6. "This decrease was less pronounced than the EGTA-induced reduction in EPSC amplitude of PhTX324 treated control synapses, indicating that control synapses are more EGTA sensitive during homeostatic plasticity than DTSpre mutant synapses". They don't use statistics to underline this statement.

- We are sorry for not having provided statistics to support this statement. A comparison between EPSC amplitudes recorded in PhTX + EGTA-treated controls and PhTX + EGTA-treated DTS mutants gave a p-value of 0.0197 (unpaired t-test), whereas there was no significant difference between both groups in the absence of EGTA treatment ($p=0.7$). We also carried out a 2-way ANOVA considering two groups (control and DTS) and two treatments (PhTX and PhTX+EGTA) followed by a multiple comparison test (Sidak's). This analysis resulted in a significant difference between the treatments in the control group (**** $p<0.0001$) and in the DTS group (** $p=0.004$). This information was now added to figure legend 5.

7. Could they plot quantal content data for figure 5?

- We did not plot quantal content in figure 5 because we did not record mEPSC amplitudes in all experimental groups shown in this figure. However, we recorded mEPSP amplitudes in all groups and did not observe significant differences between all groups (Figure S5a), implying that the observed changes in EPSC peak amplitude reflect corresponding changes in quantal content in all groups.

8. They should take reference to earlier work at the NMJ using EGTA-AM esters. They anyway strongly prefer to refer to their work and homeostatic work of the Davis lab. A more balanced referencing and discussion of also other work concerning the elucidation of presynaptic release machinery at the NMJ would be appreciated. This also applies to recent work elucidating mechanisms of loose versus tight coupling in this system.

- We added more references on presynaptic release mechanisms and coupling to the introduction and the discussion (Böhme et al., 2016; Reddy-Alla et al., 2017; Bruckner et al., 2017; Kittel et al. 2006). (for instance see page 4, line 76)

9. Could they perform RRP estimates under EGTA-AM?

- We now performed RRP estimates after EGTA-AM application (Figure S5d,e). The results are in line with our EPSC amplitude data, demonstrating that the RRP of PhTX-treated synapses is more EGTA-sensitive than in the absence of PhTX (Figure S5e). Moreover, we observed increased EGTA-sensitivity of the RRP upon presynaptic dysbindin overexpression (Figure S5e). Due to the fact that EGTA predominantly reduced EPSC amplitudes during the first third of the 60-Hz train, RRP size was analyzed with the Elmqvist and Quastel (EQ) method (Elmqvist and Quastel, 1965), which gave similar cumulative

EPSC values at non-EGTA-treated NMJs as compared to our standard analysis (back-extrapolation of cumulative EPSC amplitudes during steady-state of the train, which could not be applied because it yielded negative cumulative EPSC amplitude values; Figure 6).

10. Given that the senior author was involved in work implying Rim-binding protein in the homeostatic scaling, I think they should investigate the role of rim-bp for plasticity after proteasome perturbation.

- We analyzed rim-bp in the context of proteasome perturbation (Figure 7b). Interestingly, loss of rbp does not affect the increase in release upon proteasome inhibition.

11. usage of the term synapse: a whole NMJ terminal is "a synapse" obviously in their terminology. They should clarify this.

- Thanks for the suggestion. We now clarify this on page 2 (line 35) and state that "we use the terms 'synapse' and 'NMJ' interchangeably".

12. Dlg is not a postsynaptic DENSITY marker here but labels the whole membrane of the postsynaptic reticulum

We changed the text accordingly and used 'postsynaptic reticulum' instead of 'postsynaptic density' (page 12, line 266).

REVIEWERS' COMMENTS:

Reviewer #1 (Remarks to the Author):

The authors have executed several new experiments that have adequately addressed my concerns. In my view, the paper is now acceptable for publication in its current form.

Reviewer #2 (Remarks to the Author):

Reviewer 2:

In their rebuttal, the authors opt against radical changes in their manuscript (e.g. splitting it up). Their explanation is brief: "After adding new data and discussing this issue with the editor, we decided against splitting the manuscript". One of the main criticisms in my original report ("the [] findings in the first 6 figures remain unexplained and the most crucial questions are not addressed" and "there is little connection between the mechanisms described in mammalian and fly synapses, also little efforts to establish this, and generally there is a lack of overarching models to explain homeostatic control") remain. The authors mention some attempts to test Dunc13 mutants. It is not clear which experiments. These experiments are part of another manuscript. It is also not clear if these experiments provide a critical test to previously postulated working models for homeostatic plasticity. One new experiment on the active zone matrix protein Fife is included and may be considered an attempt to connect to previous observations on other matrix proteins (Bassoon, Piccolo), but such a connection is apparently not found. And several attempts to link with previous findings are mentioned in the rebuttal, but apparently these all failed to make such a connection. The authors decide to push this forward ("Future studies will shed light on the molecular signaling pathways relating the modulation [] vesicles with presynaptic Ca²⁺ signaling, RRP size and UPS function [] during [] homeostatic plasticity"). The bottom line is that no new experiments are provided to explain the major findings in the 1st 6 figures and this manuscript still follows the apparent tradition in this field to implicate new genes, pathways and biophysical principles in homeostatic control, but fail to build strong common ground for a mechanistic understanding of homeostatic plasticity. In the end, this is not good for the field. An overarching model is still lacking and there is little conceptual foundation for future studies in this field.

Instead the authors choose to move forward on Dysbindin and performed 3 informative sets of experiments. Especially the demonstration that Dysbindin over-expression increases RRP size (and that these extra vesicles are EGTA-sensitive) is an important addition. The authors argue against more experiments on Ca²⁺-influx. They provide in principle good arguments (evidence already available in literature), but in a field with so little connection between existing studies, it would have been so reassuring to see if conclusions from such previous papers are also true under the exact conditions of the experiments in the current manuscript (and also the question if the observed effects are indeed downstream of snapin and SNAP25, as proposed). The new experiments included in the revised manuscript describe the role of dysbindin better, but still not really thoroughly. Everything is based on physiology. An ultrastructural analysis of the synapses, as proposed in the original review report is not performed. Given the central role of different populations of vesicles and their distance to calcium channels (active zones) in the author's explanation of the Dysbindin data, this seems a logical and justified request for publication in a high impact journal.

Reviewer #4 (Remarks to the Author):

I am satisfied with their revision and recommend publication of the manuscript in its current form.

RESPONSE TO REFEREES

Wentzel *et al.*

Reviewer #1 (Remarks to the Author):

The authors have executed several new experiments that have adequately addressed my concerns. In my view, the paper is now acceptable for publication in its current form.

Reviewer #2 (Remarks to the Author):

In their rebuttal, the authors opt against radical changes in their manuscript (e.g. splitting it up). Their explanation is brief: “After adding new data and discussing this issue with the editor, we decided against splitting the manuscript”.

=> As suggested by the editor, we focused our efforts on providing more data to (i) support the specificity of proteasome perturbation, (ii) strengthen the interpretation of the EGTA data, (iii) further investigate a potential role of other presynaptic genes/proteins, and (iv) to further link *dysbindin* to the first part of the manuscript. Based on the new data and analyses, we decided against splitting the manuscript into two parts.

One of the main criticisms in my original report (“the [] findings in the first 6 figures remain unexplained and the most crucial questions are not addressed” => We concentrated on the analysis of *dysbindin* to start unraveling the molecular mechanisms underlying the findings shown in the first six figures. Specifically, we provide genetic evidence that the homeostatic plasticity gene *dysbindin* is required for release potentiation after proteasome perturbation (Figure 7). We also provide genetic evidence for links between *dysbindin*, presynaptic proteasome function and homeostatic plasticity (Figure 7). In addition, we show that *dysbindin* overexpression leads to an increase in release, EGTA-sensitivity of release, RRP size, and EGTA-sensitivity of the RRP (Figure 8 and S5). Collectively, these results indicate that *dysbindin* regulates a pool of EGTA-sensitive vesicles that is negatively regulated by proteasomal degradation under baseline conditions, and which is recruited during homeostatic plasticity. These data therefore shed light onto the mechanisms underlying the observations described in figures 1-6.

[] and “there is little connection between the mechanisms described in mammalian and fly synapses [],

=> We now dedicate a paragraph in the discussion to emphasize connections between *Drosophila* and mammalian systems (p.24, l.574). Moreover, we reference and discuss several mouse studies: For instance, we relate our findings to a recent study demonstrating accelerated release kinetics of ‘slow’ synaptic vesicles during presynaptic homeostatic plasticity at the mouse NMJ (Wang et al., 2016), or to a recent paper on pharmacological proteasome perturbation at cultured mouse hippocampal neurons (Hakim et al., 2016).

[] also little efforts to establish this, and generally there is a lack of overarching models to explain homeostatic control”) remain.

=> We are convinced that our findings significantly advance the understanding of the mechanisms underlying presynaptic homeostatic plasticity. However, we are far away from ‘overarching models of homeostatic control’.

The authors mention some attempts to test *Dunc13* mutants. It is not clear which experiments.

=> As mentioned in the last response letter, we tested homeostatic plasticity in recently published *dunc13A* and *dunc13B* mutants (Böhme et al., 2016). We observed a pronounced increase in quantal content upon PhTX application in *dunc13A* and *dunc13B* mutants, suggesting that the acute induction of homeostatic potentiation of release does not require these genes. As the focus of the current manuscript is to establish links between homeostatic plasticity and proteasome function, we decided against further investigating *dunc13*.

These experiments are part of another manuscript. It is also not clear if these experiments provide a critical test to previously postulated working models for homeostatic plasticity.

=> We studied the role of *dunc13A* in release depression and homeostatic plasticity in the context of another manuscript that aims at understanding the interplay between different homeostatic plasticity forms. This manuscript contains the data on *dunc13A* in PhTX-induced homeostatic plasticity mentioned above.

One new experiment on the active zone matrix protein Fife is included and may be considered an attempt to connect to previous observations on other matrix proteins (Bassoon, Piccolo), but such a connection is apparently not found.

=> It is worth highlighting that we investigated nine mutants, including the three active zone-related genes (*rim*, *rbp* and *rab3-GAP*), with a defect in homeostatic plasticity in the context of proteasome function. Our results demonstrate that specific homeostatic plasticity genes are required for release potentiation upon proteasome perturbation.

And several attempts to link with previous findings are mentioned in the rebuttal, but apparently these all failed to make such a connection. The authors decide to push this forward (“Future studies will shed light on the molecular signaling pathways relating the modulation [] vesicles with presynaptic Ca²⁺ signaling, RRP size and UPS function [] during [] homeostatic plasticity”). The bottom line is that no new experiments are provided to explain the major findings in the 1st 6 figures and this manuscript []

=> As mentioned above (point 2), the results of our new experiments further link the homeostatic plasticity gene *dysbindin* to release modulation by proteasome function.

[] still follows the apparent tradition in this field to implicate new genes, pathways and biophysical principles in homeostatic control, but fail to build strong common ground for a mechanistic understanding of homeostatic plasticity. In the end, this is not good for the field. An overarching model is still lacking and there is little conceptual foundation for future studies in this field. Instead the authors choose to move forward on Dysbindin and performed 3 informative sets of experiments. Especially the demonstration that Dysbindin over-expression increases RRP size (and that these extra vesicles are EGTA-sensitive) is an important addition. The authors argue against more experiments on Ca²⁺-influx. They provide in principle good arguments (evidence already available in literature), but in a field with so little connection between existing studies, it would have been so reassuring to see if conclusions from such previous papers are also true under the exact conditions of the experiments in the current manuscript (and also the question if the observed effects are indeed downstream of snapin and SNAP25, as proposed).

=> We decided against repeating Ca²⁺-imaging experiments because of the reasons mentioned in the last response letter. Furthermore, it is worth noting that four independent Ca²⁺-imaging data sets of four studies (Müller and Davis, 2012; Younger et al., 2013; Wang et al., 2014, Müller et al., 2015) indicate a role for presynaptic Ca²⁺ signaling in homeostatic plasticity. Instead of confirming genetic links between *SNAP25*, *snapin* and *dysbindin* described in Dickman et al. (2012), we focused on experiments to directly link *dysbindin* to proteasome function and homeostatic plasticity.

The new experiments included in the revised manuscript describe the role of dysbindin better, but still not really thoroughly. Everything is based on physiology. An ultrastructural analysis of the synapses, as proposed in the original review report is not performed. Given the central role of different populations of vesicles and their distance to calcium channels (active zones) in the author's explanation of the Dysbindin data, this seems a logical and justified request for publication in a high impact journal.

=> This is a good suggestion. Unfortunately, we were not set up to realize EM-based ultrastructural analysis during the resubmission period.

Reviewer #4 (Remarks to the Author):

I am satisfied with their revision and recommend publication of the manuscript in its current form.